# Effect of discontinuous fair-share emissions allocations immediately based on equity

Yann Robiou du Pont [1,2] ✉, Mark Dekker [1,3], Detlef van Vuuren[1,3] & Michiel Schaeffer[1,4,5]

National emissions targets are collectively insufficient to align with the Paris Agreement. The fair-share literature assesses whether these targets are fair and ambitious in comparison to emissions trajectories based on equity principles. Such emissions trajectories commonly start at present-day emissions levels. Here we show that these continuous trajectories inherently reward past inaction and increasingly do so with their iterative updates. We provide an approach to allocating emissions trajectories based on equity principles applied with immediate effect. The resulting discontinuous national trajectories not starting at current emissions levels imply significant immediate international support to fund rapid mitigation globally. Modelling allocations with or without continuity has remarkable consequences for the relative implied contributions to international support among high-income countries. We find that emissions targets of G7 countries, Russia and China are responsible for most of the global 2030 ambition gap, while only some countries align with their 1.5 °C allocation.

The Global Stocktake under the United Nations Framework Convention on Climate Change (UNFCCC) found in 2023 that the aggregated impact of the 2030 countries' pledges is insufficient to achieve the goals of the Paris Agreement[1]. To inform negotiations on the periodic improvement of national emissions pledges, studies assess their ambition against quantifications of each country's fair share of the remaining global emissions space to limit global warming to 1.5 °C and well below 2 °C in line with the Paris Agreement[2–6]. The literature on fair emissions levels can contribute to explaining, or even enhancing, the ambition of national pledges[7–10] that insufficiently describe how they are 'fair and ambitious'[11] as required under the Paris Agreement. Governments can legislate to adopt emissions targets recommended by independent bodies based on equity-based literature[8–10,12]. Additionally, this fair-share literature designed to inform international negotiations on fair contributions to achieve the Paris Agreement is also used by courts of law to determine the adequacy of countries' emissions objectives under national laws[6,13–17]. Recent literature has compared the ambition of Nationally Determined Contributions (NDCs) to possible emissions allocations based on fairness principles[2–6,18,19]. However, most approaches allocate continuous emissions trajectories to countries, starting at their current emissions levels[6,20]. Such a modeling choice of continuous allocations favors, in the near term, countries with high current emissions resulting from relatively minor past efforts to reduce emissions[6]. This influence of present-day emissions on near-term emissions allocations also affects the ambition assessment of NDCs. The choice to model continuous emissions trajectories leads to a more lenient ambition assessment of NDCs for countries with higher than equitable emissions, to the disadvantage of others. Successive future literature updates of continuous emissions trajectories would increase this legacy effect as we approach the target dates, currently 2030.

Here we show how allocating discontinuous emissions trajectories solves the near-term legacy influence present in the fair share literature, regardless of the equity principle modeled. We quantify two

[1]Copernicus Institute of Sustainable Development, Utrecht University, Princetonlaan 8a, Utrecht, Netherlands. [2]Centre for Climate and Energy Transformation (CET), The University of Bergen, Christies gate 18, Bergen, Norway. [3]PBL Netherlands Environmental Assessment Agency, Bezuidenhoutseweg 30, The Hague, Netherlands. [4]Climate Analytics, Ritterstraße 3, Berlin, Germany. [5]Universitas Islam Internasional Indonesia, Jalan Raya Bogor KM. 33.5 Cisalak, Sukmajaya Depok, West Java, Indonesia. ✉e-mail: yann.rdp@climate-energy-college.org

methods to allocate discontinuous emissions trajectories, starting at emissions levels only based on fairness criteria, rather than present-day observed emissions levels. Here, 'fair' or 'equitable' allocations do not seek to reflect the personal view of any authors, but emissions allocations resulting from effort-sharing approaches based on principles of distributive justice and the Common But Differentiated Responsibilities and Respective Capabilities (CBDR-RC) of the Framework Convention and Paris Agreement[1]. We apply these methods to allocate to countries the emissions of a range of global scenarios with warming ranging from 1.5 °C to over 4 °C and compare the resulting allocations to the countries' NDCs. We discuss the emissions gaps between these allocations and their pledges. Finally, we discuss the results of these discontinuous trajectories with results from the recent literature.

## Results

### Avoiding a grandfathering influence from continuous fair share trajectories

The literature quantifying emissions allocations based on diverse principles of distributive justice, including the Paris Agreement's CBDR-RC, agrees on the insufficiency of the NDCs of most of the largest emitting countries[2,3,5,6,18–21]. Although there are divergences on the modeling choices of equity concepts[21,22], this literature focused on a 'continuous' allocation of emissions trajectories starting at current emissions levels. In this context, 'continuous' refers to trajectories starting at current emissions levels, rather than immediately at equitable levels. Effort-sharing formulas can achieve such continuity by design[2,3,23] or through a transition period added to ensure continuity[3,18,20] towards allocations only based on fairness considerations. This continuity is also commonly assumed when allocating national carbon budgets over time into emissions trajectories[18,24,25].

The legacy influence of current emissions levels on near-term emissions allocations is described here as a 'grandfathering' effect[6]. This grandfathering influence on equity-based emissions allocation is strongest in the near term and increasingly affects the ambition assessment of NDCs in 2030. As we near 2030, a given NDC's emissions target will be closer and closer to a continuous emissions allocation that is iteratively updated (Fig. 1). The grandfathering allocation is criticized for its lack of ethical basis[26–28] and has been shown to penalize the poorest countries[20] as it preserves a status-quo, including current inequalities. Prior to the Paris Agreement, a study highlighted the value of a moderate grandfathering[29], from a political theory perspective, with a realist justification for negotiations and a utilitarian justification. Indeed, the pledges of many high-emitters only align with a grandfathering allocation[3]. However, the IPCC has highlighted the need for a fair distribution of mitigation efforts, excluding grandfathering, in order to achieve an effective global agreement on emissions reductions[21,27]. Likewise, recent reports of scientific advisory bodies have disapplied grandfathering when presenting fair-share emissions allocation[12,30]. The Paris Agreement now requires NDCs of the highest possible ambition that reflect equity. A recent study[6] described grandfathering allocations as not in line with international law. It identified that all continuous allocations entail elements of grandfathering but did not offer a solution.

The key motivation for allocating continuous emissions scenarios is to address the need for emissions trajectories that countries can implement domestically[27,29]. However, different from emissions scenarios from Integrated Assessment Models (IAM), equity allocations do not engage with feasibility concerns but solely focus on effort allocation irrespective of where emissions reductions take place. Past delays of emissions reduction progressively leads to steeper fair share trajectories that are unlikely to be considered politically, technically or economically realistic for any country[27]. Recent studies on budget allocations have shown already depleted budgets for high-emitting countries[24,31]. Modeling trajectories from current levels in such instances leads to countries accumulating further excess depletion to be compensated by substantial negative emissions in the future. However, equity-based allocations serve as a proxy to distribute mitigation efforts and do not need to be met through domestic mitigation exclusively. Instead, countries can achieve their equity-based

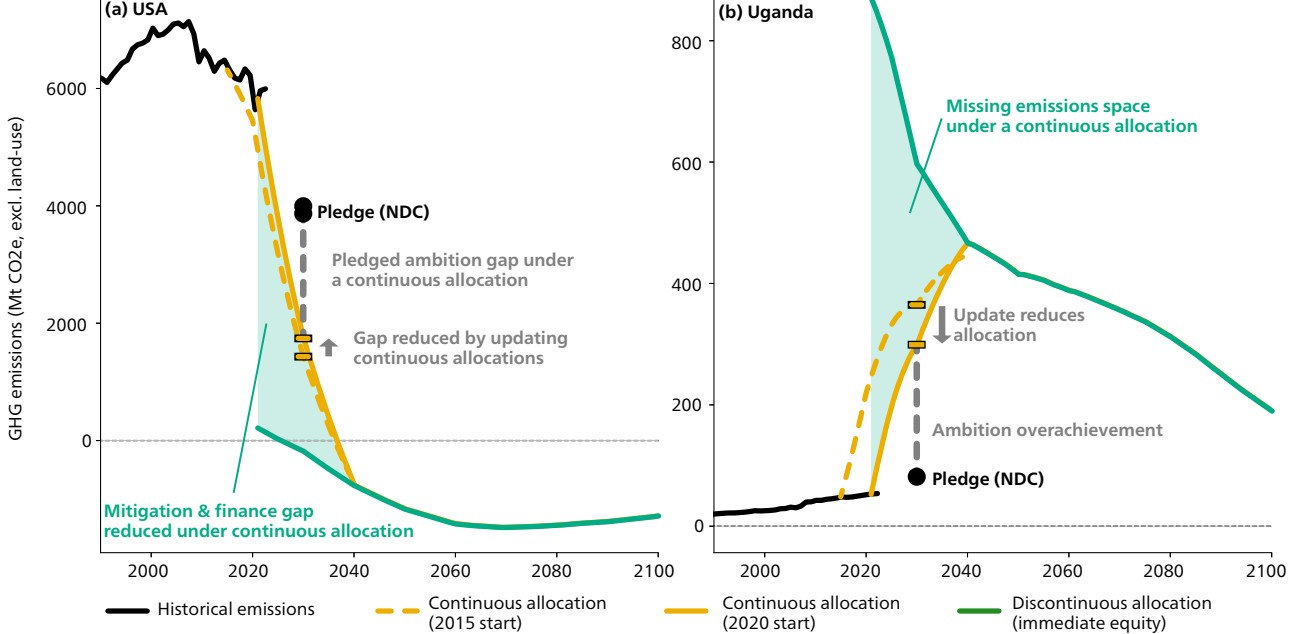

**Fig. 1 | Comparison of emissions allocations when modeling continuous and discontinuous allocations taking the USA. a** and Uganda (**b**) as examples. The successive update of continuous allocations emissions allocations, here illustrated with the addition of a transition period starting 2015 (dashed red) and 2020 (red), rewards countries with insufficient Nationally Determined Contributions (NDCs) by decreasing the 2030 gap (gray dashed line) with the emissions allocation. Cumulatively over the transition period, the allocation difference with a discontinuous allocation (green area), here based on Approach 2, affects the additional effort, and possible climate finance, needed to align NDC with fair allocations.

emissions allocations through a combination of domestic effort and international cooperation[27,32]. As such, emissions allocations do not require to be continuous, and countries can immediately provide an equitable share of the global mitigation effort, beyond the limitations of what is feasible to implement within their borders. Immediately contributing to mitigation efforts outside of their own territories would be better able to remedy current inequities caused by past emissions as well as address the current emissions gap between current NDCs and the Paris Agreement targets.

To inform this extraterritorial contribution, the IPCC sixth assessment report calls for research extending equity frameworks to quantify equitable international support as the difference between equity-based national emissions scenarios and national domestic emissions scenarios[22]. Such frameworks would enable assessing the ambition of countries' total contribution to global mitigation through domestic and international cooperation jointly. International cooperation to mitigate global emissions through bilateral agreements (as in the Swiss NDC), financial support or trading of Internationally Transferred Mitigation Outcomes (ITMOs), is now facilitated with the agreement of trading rules under Article 6 of the Paris Agreement at UNFCCC COP29. Such mechanisms offer a solution to progress towards an equitable distribution of the mitigation effort while contributing to the funding of mitigation measures, in line with cost-optimal implementation of mitigation measures across countries. The utilitarian justification[29] for a moderate grandfathering relies domestic mitigation costs and is no longer relevant when allocations can be traded to achieve a globally cost-effective pathway[27].

Emissions scenarios from IAM assume the implementation of the cheapest mitigation option in each region without specifying which country should fund these measures. Implementing these scenarios without international cooperation – assuming that each country should fund the measures modeled for their territory – would represent a much greater fraction of Gross Domestic Product (GDP) in regions with lower GDP per capita[33]. While much of the mitigation potential is in countries with low GDP per capita, countries with the greatest financial capacity fail to provide sufficient unconditional finance[34,35] or meet their own pledges[36]. Additionally, accounting for the higher cost of access to capital in poorer countries would impact mitigation costs and imply greater domestic mitigation in richer countries than currently found in IAMs[37–39]. The implementation of the Paris goals, or of IAM scenarios, thus requires significant and immediate international support[35,40,41]. Equitably implementing a global IAM trajectory can imply that countries' domestic emissions trajectories follow cost-optimal scenarios – that can be downscaled at the national level[42] – and provide (or receive) the climate finance needed to mitigate overseas the difference with their equitable allocation. Compared to the domestic measures aligned with IAM trajectories, countries could pursue greater domestic mitigation to reap important co-benefits not accounted for, which can cover a substantial share of the mitigation costs[43]. In practice, countries with high responsibility and capability can align with the Paris Agreement with an NDC within their fair shares, met through a combination of domestic mitigation of the highest possible ambition[44], and funding to mitigate global emissions overseas[12]. As a novel mechanism, the international trading of mitigation outcomes raises implementation issues regarding the additionality of the finance and of the funded mitigation measures. Scrutiny will be needed to ensure the integrity of mitigation measures under Article 6 whose implementation rules were just adopted at COP29, with safeguards on human rights and the additionality of emissions reductions[41,45,46].

## Quantification discontinuous emissions trajectories

Here we quantify two sets of emissions trajectories immediately based on equity principles and that do not start at current emissions levels (see "Methods"). The two methods combine the equity principles of capability and responsibility[21] to reflect the principles of the UNFCCC

and the Paris Agreement, notably CBDR-RC. The literature suggests several approaches, conceptual or statistical, for combining different equity principles into a single allocation method (see Discussion). Here we apply each of the two equity principles to allocate global positive or negative emissions separately. This differentiated treatment of negative emissions extends a study from Fyson et al.[47] that allocated negative emissions only, based on responsibility or capability. Fyson et al.[47] explain that obligations to deliver negative emissions require uncertain technologies made necessary because of insufficient global emissions reductions to date. That study alone could not be used to inform economy-wide emissions targets, and thus not assess the ambition of NDCs, as it only allocated negative emissions and assumed that positive emissions follow least-cost pathways (that is, no equity principle is applied to gross emissions)[47]. Here, Approach 1 first allocates global negative emissions across countries based on their capability, assessed through GDP per capita, and then allocates global positive emissions to equalize historical responsibilities over the total net emissions (positive + negative, see "Methods"). Under this approach, rich countries are required to fund most of the negative emissions that require important research and development costs without local co-benefits[47]. Approach 1 also ensures equal cumulative per capita emissions over the 1990-2100 period. The present results are therefore favorable for high historical emitters compared to accounting historical emissions since 1950 (Supplementary Data). Under Approach 2, all countries contribute to positive emissions reductions based on their wealth. Negative emissions, needed because of the world's important historical emissions, are then allocated proportionally to countries' individual historical responsibilities. There, countries' cumulative emissions allocations are not only based on their historical responsibility. Looking at the global emissions scenarios, the positive emissions refer here to the projected physical emissions (e.g., fossil fuels, agriculture). The negative emissions here refer to emissions captured through Carbon Dioxide Removal, excluding those from Direct Air Capture and Land Use, Land-Use Change, and Forestry (LULUCF) unlike Fyson et al.[47].

In addition to representing CBDR-RC, the modeling of responsibility and capability also reflects considerations present in national policies. The European Union (EU) used a capability approach, based on GDP per capita, to allocate across its member states the mitigation effort of its first NDC target[48,49] and to negotiate effort-sharing under the new Fit for 55 package[50]. However, this capability criterion is not used to determine the emissions objectives of the EU NDC itself, which are not based or justified by equity considerations applicable to all[32]. The capability principle reflects notions of progressive income taxation that many countries have implemented[51]. Responsibility can be related to the 'polluter-pays' principle that many countries recognized in the Rio 1992 declaration and use in national law.

Here, countries' responsibility is studied solely based on territorial emissions accounted under UN frameworks. Other emissions frameworks account for emissions linked to consumption, fossil fuel extraction, or carbon intensity of countries' income[52]. Such accounting could lead to more stringent allocations for countries with higher responsibilities, compared to territorial emissions, regarding their consumption footprint (the EU, Switzerland, Japan, Singapore), income footprint (Norway, Switzerland, Saudi Arabia, Australia) and extraction-based emissions (Canada, Saudi Arabia, Norway, Australia)[52]. Importantly, top-down effort-sharing formulas, such as those used here, may lose relevance for countries with very small and isolated populations (e.g., small island states). Such countries may have limited technical options to mitigate emissions and limited access to some options given the small size of their economies.

## Fair-share allocations

Under both approaches, emissions allocations start at levels that only depend on the global emissions scenario and countries' historical

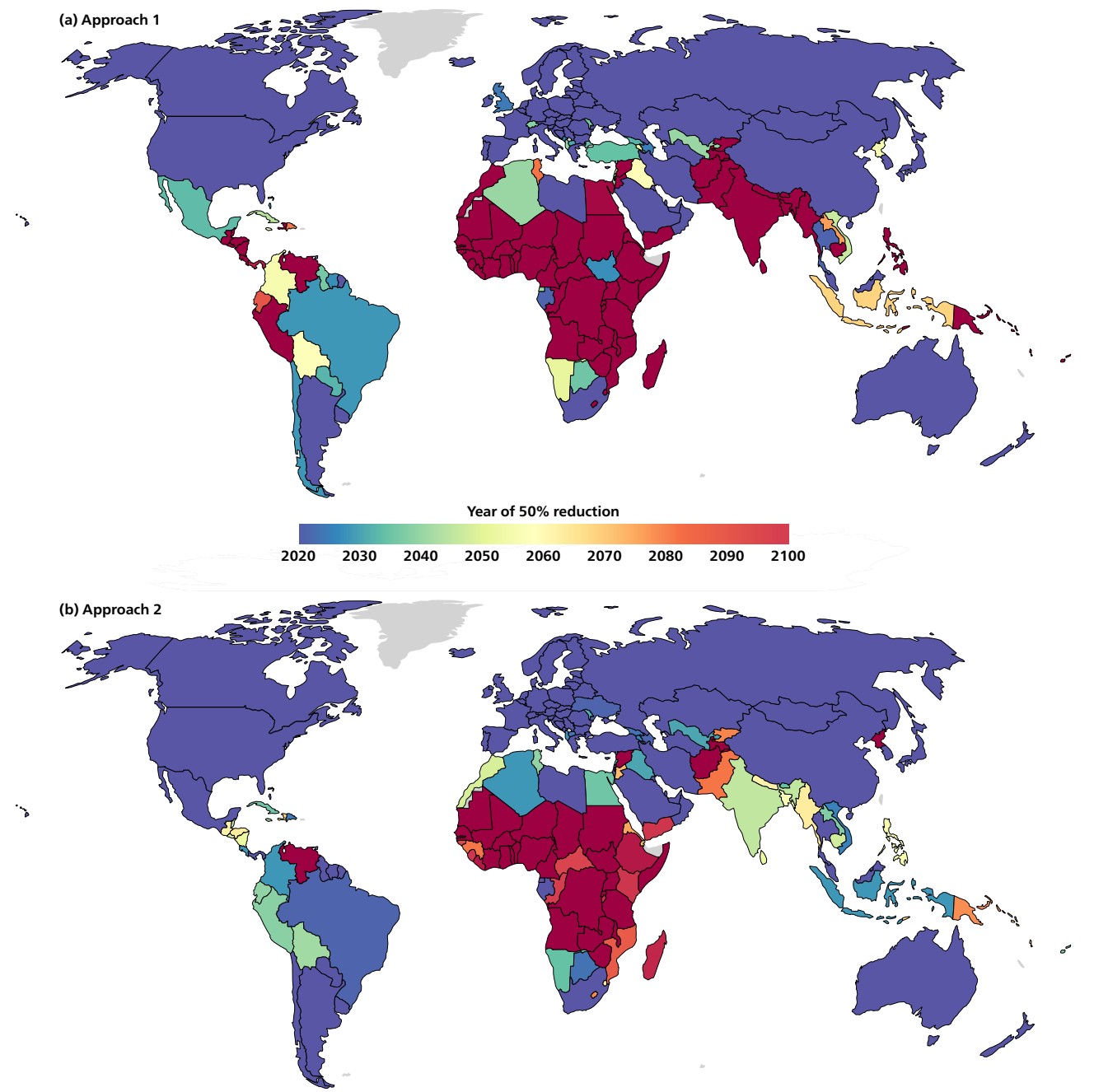

**Fig. 2 | Dates when emissions allocations first reach 50% below 2020 under Approach 1. a** and Approach 2 (**b**), based on 1.5 °C scenarios with no or limited overshoot (C1 category).

responsibilities and capabilities (Fig. 1, see country-level results in SI). Over time, emissions allocations for all countries follow the trends of the underlying global scenario (methods), a rapid decrease and plateau in the second half of the century. The USA, Canada and Australia have immediate negative allocations before 2035 under both approaches. Approach 2 is less stringent than Approach 1 for low-income countries[53] (sub-Saharan African countries) and for countries with high historical responsibility (USA, Russia, Qatar and other fossil fuel extracting countries, see Fig. 2 and Supplementary Data). These different stringencies of emissions allocations do not always change the warming assessment of countries' NDCs (Fig. 3). Emissions objectives, such as NDCs or net-zero targets, should only be considered aligned with the present allocations if earlier emissions match the discontinuous allocation as well, which implies immediate contribution to global mitigation. The near-term allocation of some countries, mostly

sub-Saharan countries, may exceed their current emissions and business-as-usual trajectory beyond 2030, implying mitigation efforts only later[18]. However, staying within such decreasing allocations beyond 2030 implies immediate investments, possibly with international support. International support can enable recipient countries to implement mitigation measures in line with the underlying global socio-economic scenario in the near term. Approach 2 uses allocations inversely proportional to GDP per capita[3,54] (see "methods"), resulting in high emissions allocations compared to current emissions and allocations based on business-as-usual trajectories[2,18] for low-income countries (e.g., Ethiopia, Democratic Republic of Congo). These allocations theoretically imply financial transfers that may go beyond needs-based considerations and contribute to poverty reduction through climate action[55]. The absence of continuity criteria highlights important sensitivities in emissions allocation across equity-based

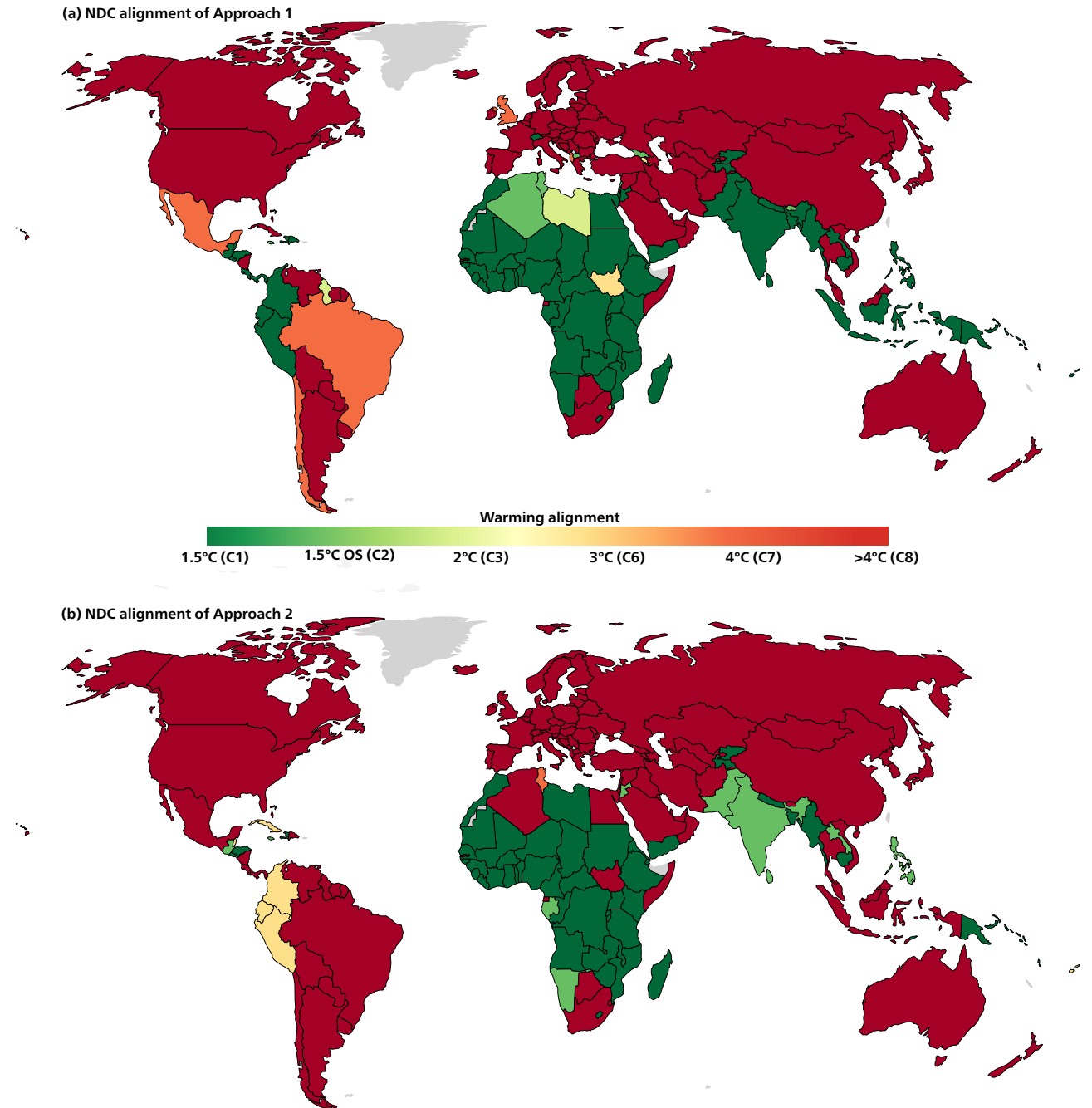

**(a) NDC alignment of Approach 1**

**(b) NDC alignment of Approach 2**

**Warming alignment**

| 1.5°C (C1) | 1.5°C OS (C2) | 2°C (C3) | 3°C (C6) | 4°C (C7) | >4°C (C8) |

**Fig. 3 | Warming assessment of Nationally Determined Contributions (NDCs) based on Approach 1. a** and 2 (**b**) allocations of global emissions scenarios limiting global warming to 1.5 °C with no or limited overshoot (C1 category), 1.5°C with high overshoot (OS, C2), likely below 2 °C, below 3 °C or below 4 °C warmings. Colors at the edges of the legend range can reflect values outside the range, either more ambitious than a 1.5 °C allocation or less ambitious than a 4 °C allocation. NDCs should only be considered aligned with the present allocations if earlier emissions also match the discontinuous allocation, which implies immediate support and cooperative approaches.

formulas that are otherwise dampened by the need for continuity in the near term and the declining global emissions space in the longer term.

The effect on the countries' near-term tradeable emissions of adding a transition phase to ensure the continuity of an emissions allocation schematized in Fig. 1. As highlighted in previous studies[14], adding a transition period greatly influences near-term allocation through a grandfathering influence. Here we show an additional effect of continuous allocation where their updates reward inaction by closing the ambition gap between the updated equity-based allocations and an insufficient NDC (exemplified in Fig. 1).

Looking at the geography of reduction rates, Fig. 2 shows the date when allocations are half of 2020 levels based on a 1.5 °C trajectory with no or limited overshoot (C1 scenarios, excluding bunkers and LULUCF emissions; see "Methods"). Under both approaches 1 and 2, the allocations of most countries reach half of 2020-levels by 2030. The allocations of some sub-Saharan and South Asian countries remain positive over the century.

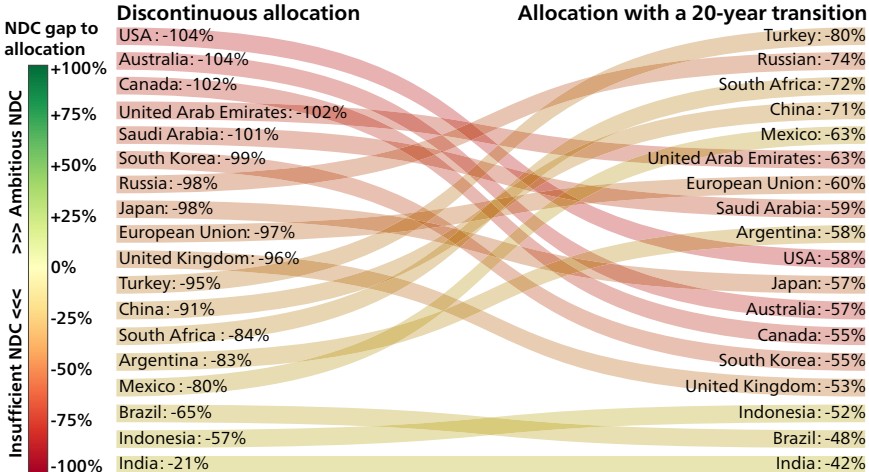

**Fig. 4 | Effect on the 2030 emissions gap between countries Nationally Determined Contributions (NDCs) and their 1.5 °C allocations of adding a 20-year transition period from current emissions levels (right side), compared to allocations calculated in Approach 2 (left side).** The ranking of additional mitigation effort and potential international support needed for countries to align their NDC with their allocation changes when using a transition period. Results are shown for G20 countries with NDCs above their allocation. The color legend is based on the discontinuous allocation.

The 'warming maps' (Fig. 3) show the warming alignment of countries' NDCs, that is, the warming associated with the most ambitious global scenario underlying the allocation that is above their NDC in 2030. This 'warming alignment' of a country also reflects the expected global warming level when all other countries follow a similar level of ambition. The differences in terms of emissions allocations across the two approaches do not translate into important differences for countries' ambition assessments. Most countries have an ambition assessment either 1.5 °C-aligned or not even 4 °C-aligned. The main reason is that the effect of the current inequities across countries' 2030 allocations overwhelms the relative spread in numerical targets across countries. This polarization of results reflects the extreme disparities of the current situation, considering countries' responsibilities and capabilities. This effect increases as we delay climate action and as we near 2030. Many countries have committed to NDCs much more ambitious than their 1.5 °C-allocations, mostly sub-Saharan countries. Countries with NDCs within their fair-shares could sell emissions space (possibly through conditional NDCs), possibly under Article 6 of the Paris Agreement, which could fund the implementation of the mitigation measures implied by the cost-optimal scenarios[42]. Most high-income countries have largely insufficient NDCs, even when historical emissions accounted only since 1990. Approach 2 also yields a more stringent assessment for the NDCs of the UK, Switzerland and multiple countries, mostly in Latin America and South Asia.

The absence of continuity also changes the ranking of countries in terms of additional mitigation effort needed to align with their allocation, which potentially affects their share of the climate finance to be provided globally. Figure 4 provides an illustrative case of the influence of adding a 20-year transition period on the gap between the emissions allocations of G20 countries and their respective NDCs. In theory, each country's relative contribution to total international climate finance can be proportional to how much each country's NDC deviates from its fair-share trajectories. Compared to a traditional continuous approach, applying a discontinuous approach implies here a much higher obligation to contribute to global mitigation and possibly international finance for all G20 countries except India. In terms of the ranking of the emissions gap between NDC and allocation, assuming a transition period benefits Canada and Australia (moving down 9 positions), the USA and South Korea (each 8 positions). This shows that continuous pathways reward such countries for their history of comparably low mitigation efforts, lowering their

implied contribution to international climate finance. Other countries, including China, Türkiye, South Africa and even the EU move down in the ranking of the ambition gaps when removing the transition period.

Discussions about climate finance in the context of the UNFCCC have often referred to the perceived obligations of, for example, high versus low-income countries. The comparison between continuous and discontinuous fairness allocations highlights the difference amongst high-income countries when considering the amount of finance needed to meet their fair shares.

In addition to capability and responsibility, equality is the third equity principle described in the IPCC AR5[21,27]. IPCC reports do not present equity-based emissions allocations since AR5, despite available studies and its importance for courts of law[13]. The egalitarian approach modeled as equal per capita emissions is not explicitly mentioned in the Paris Agreement or international environmental law[6] but it can reveal the inequalities of emissions spaces claimed through NDCs. Figure 5 shows the equal per capita allocation where each country's share of global emissions is proportional to its population projection at every point in time[18]. Even discarding countries' CBDR-RC, as modeled in approaches 1 and 2, this equality-based assessment yields similar warming assessments for most large emitters. In other words, the NDCs of many sub-Saharan countries are below the equal per capita levels of a 1.5 °C scenario. However, the NDCs of many countries with high responsibility and capability, which are meant to reflect their 'highest possible ambition'[1,44] and account for CBDR-RC do not even reflect an equal per capita share of a business-as-usual trajectory, itself yielding warming impacts that hit some countries much harder than others. Establishing a country's alignment with the CBDR-RC principle depends on the methods used to quantify the responsibility and capability principles[18,24]. However, the misalignment of an emissions target with a simple equal per capita allocation can be used to characterize a misalignment with the CBDR-RC principle, for countries with higher-than-average capability and responsibility. In a recent ruling[15], the European Court of Human Rights used a simple equal per capita allocation to comment on Switzerland's inadequate emissions levels while recognizing the need to account for its CBDR-RC.

## Discussion
Here we discuss how this study's modeling choices compare to the literature regarding the continuity assumption and the combination of equity principles. Then, we compare results.

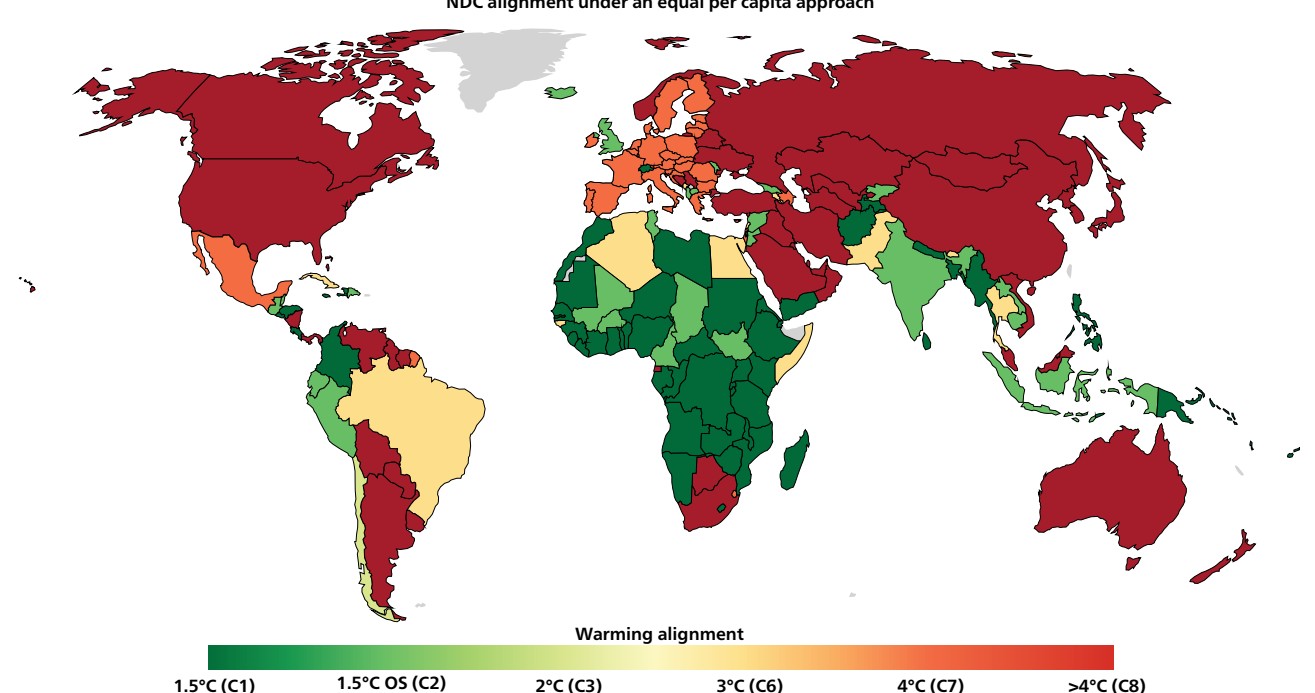

**Fig. 5 | Warming assessment of Nationally Determined Contributions (NDCs) based on an equal per capita allocation of global emissions scenarios limiting global warming to 1.5 °C with no or limited overshoot (C1 category), 1.5 °C with high overshoot (OS, C2), likely below 2 °C, below 3 °C or below 4 °C warmings. Colors at the edges of the legend range can reflect values outside the range, either more** ambitious than a 1.5 °C allocation or less ambitious than a 4 °C allocation. NDCs should only be considered aligned with the present allocations if earlier emissions also match the discontinuous allocation, which implies immediate support and cooperative approaches.

A study[28] called for transparency regarding ethical choices behind quantifications of fair contributions under the Paris Agreement. It categorized existing literature based on whether a grandfathering perspective or approach had been used explicitly or through the addition of a transition period. Our analysis reveals a broader form of grandfathering influence not previously identified, resulting from the choice of modeling continuous emissions allocation. This modeling choice is not necessary to allocate global mitigation efforts and is at odds with legal and ethical principles. Greater transparency is needed in future studies regarding the motivations and implications of modeling continuous allocations. A transparent disclosure on the inclusion or not of a continuity criterion in models can also inform courts on how the modeling assumptions affect the assessment of whether or not a country's target is legally sufficient[6,28].

Even models that do not have a transition period but rely on continuous allocations have that grandfathering influence. In the model behind the Climate Equity Reference Calculator[2,23] (CERC), the continuity of the emissions trajectory results from the choice of allocating the mitigation burden to depart from a reference trajectory rather than allocating the remaining emissions space[2]. Such an approach based on reference trajectories can be adequate to assess the ambition of an emissions target when provided with a corresponding reference scenario, e.g. pledges adopted in 2015 can be compared to allocations starting in 2015. The grandfathering effect could occur when the ambition of an older target is assessed against a newer allocation based on a newer reference scenario that no longer reflects the effort that the target represented when adopted. With successive updates, the allocated emissions trajectory at a given year would come closer to even a business-as-usual target. An allocation starting the year prior to the target will find it to be close to equitable levels. Therefore, such an approach may not be suited as a single metric to assess the ambition of targets adopted at different times by different countries[2,56]. To avoid the

grandfathering influence of allocation updates, it is possible not to update allocations (e.g. after the Paris Agreement), but it misses the effect of later emissions trends of countries on allocations[57]. The CERC approach can be used to determine the financial support that countries should receive to depart from a reference scenario. Because of the reliance on a reference scenario, the tradeable mitigation outcome starts at zero. This could disserve recipient countries in the near term, compared to using the allocations derived in the present study, unless finance is mobilized immediately to fund future mitigation gap.

The near-term grandfathering influence may increase the overshoot of a country's emissions budget to be compensated through lower or negative allocations later in the century[2,3,5,12,18,24]. Relying on the assumptions of future compensations reduces near term efforts and complicates accountability. Some studies allocate emissions budgets, which are not defined over time and do not have a grandfathering influence. The flexibility provided by carbon budgets over emissions pathways[18] theoretically allows countries to use their budget mostly in the near-term and justify insufficient emissions objectives, which raises issues of intergenerational justice[12]. However, budgets are not adequate to assess the ambition of emissions reduction targets without additional assumptions regarding their use over time[24]. The grandfathering influence is introduced when hypothesizing the use of the emissions budget through a continuous emissions pathway[25,58,59]. A recent study[31] suggests that countries should compensate through negative emissions for their 'carbon debt'[60,61] calculated as the observed or pledged overshoot of fair shares of the nearly exhausted 1.5 °C global carbon budget. This budget-based approach has been used to establish a breach of a legal obligation of states by courts[15,17]. However, emissions budgets are less suitable to determine minimum near term targets, which may be enforced by courts[62], and can be complemented by discontinuous pathways such as modeled in this study.

A key challenge in the fair-share literature is the combination of different equity dimensions into a single emissions allocation method[63]. Here, the responsibility and capability principles are combined to drive the allocation of positive and negative global emissions distinctly, but other approaches to combining different equity principles exist in the literature. Simply presenting emissions allocation side by side[3,5,18,64] may lead to the erroneous interpretation that the underlying principles are equivalently equitable, especially if compared to a grandfathering allocation[20]. Averaging[2,65] or applying weighting factors[2] to allocations based on different equity formulas reflects a numerical compromise rather than a multidimensional vision of equity[66]. Likewise, the online tool Climate Action Tracker (CAT) assesses and suggests targets based on an aggregation of fair-share studies[19]. These underlying studies may have mutually inconsistent hypotheses, including different starting dates that strongly influence near-term emissions allocations, different pursued warming outcomes based on different global scenarios, and budgets used differently over time. The emissions allocations from various studies are then harmonized before being aggregated. The iterative aggregations of continuous emissions allocation can increasingly impact the ambition assessment of countries' NDCs, in favor of parties with emissions above their allocations[6,28]. Alternatively, other studies apply distinct equity principles to each country[4,67,68], possibly reflecting a self-differentiated approach of equity reflective of the Paris Agreement bottom-up architecture[4] rather than applying a single principle to all countries. Under this differentiated combination, each country's fair-share formula does not affect how the share of another country is calculated. Instead, it affects the global goal applied to all countries' formulas. While these combinations of equity principles often rely on underlying continuous allocations, these are not a methodological requirement. The combination modeled in the present paper is also conceptual rather than numerical and avoids giving relative arbitrary weights to equity principles. The two modeled approaches illustrate the sensitivity of allocations to alternative combinations of capability and responsibility in the absence of continuity criteria.

The comparison of these results with the literature shows broad agreement on the insufficiency of some NDCs under the 1.5 °C goal but divergences on the ambition gaps across countries. Here, only substantial improvement, including through international cooperation, can improve the warming assessment of many NDCs. Compared to a previous warming assessment[4] (visible on Paris-Equity-Check.org) where each country follows the least stringent of three equity principles (capability, responsibility, and equality), the present approaches find NDCs to be more ambitious (1.5 °C aligned) for a few countries (including India, Indonesia, and North Africa depending on the approach) and less ambitious for countries in most Latin American and Annex I countries. Both Approach 1 and 2 also have more polarized results with fewer NDCs between 1.5 °C and 3 °C, partly because the previous warming assessment[4] relies on continuous approaches[3,69]. Compared to the CAT's 'fair shares'—the part of the CAT[19] assessments that serves to assess the ambition of the overall mitigation effort in countries' NDCs—results of the present study are more stringent for G7 countries, China and South Africa, and less for India. The CERC[23] was used to published an NDC assessment[2] in 2017 that found China's NDC to be nearly 1.5 °C aligned, unlike most other assessments[3,4,18,19], including from Chinese institutions[5]. In terms of emissions allocations, the CERC finds much higher emissions allocations for Middle Eastern fossil-fuel-exporting countries (Saudi Arabia, United Arab Emirates, Qatar) implying a right to international climate support—despite their high GDP per capita. Under its default setup (responsibility since 1950, but more settings are available), the CERC is more stringent for the EU with net-zero dates before 2030 (before 2035 in our study with responsibility since 1950), India, and the Least Developed Countries

(LDCs) collectively, since their allocations are capped by their reference scenario.

In conclusion, this study highlights how iterative updates of ambition assessments that are based on continuous emissions allocations reward a lack of action and reduce the implied provision of international support. As a result, the ratcheting-up mechanism of the Paris Agreement is currently informed by ambition assessments that iteratively ratchet down the importance of equity considerations in the near term. We offer a solution to this near-term grandfathering influence, applicable to any equity-based emissions allocation, by modeling discontinuous emissions trajectories starting immediately at equity-based levels. Applying this method, we model two approaches that reflect countries' responsibilities and capabilities to derive emissions trajectories immediately consistent with the Paris Agreement's equity principle. With this removal of an important near-term grandfathering effect, we quantify ambition gaps between countries' allocated trajectories and current or pledged emissions. As NDCs for 2035 are expected in 2025, this study shows that meeting allocations immediately based on equity requires a faster scale-up of mitigation measures along with immediate international support. Future studies can quantify the equitable international support needed for countries to align with discontinuous emissions allocations.

## Methods
### Allocations methods
**Approach 1** combines a capability-driven allocation of global negative emissions with a historical responsibility-driven allocation of global positive emissions to correct for historical responsibility and equalize per capita emissions rights over the considered period. The 'historical' period to account for responsibility here starts in 1950 to reflect recent historical emissions (Supplementary Data) or in 1990 to account for observed emissions since the first IPCC report (main text). Here, the capability allocation affects the distribution of emissions over time, but not the total budget. Richer countries then have more important negative allocations in the future and less stringent allocations in the near term. The capability-driven allocation of growing global negative emissions (under the most ambitious global pathways) requires greater negative emissions from richer countries, mostly occurring after 2030. Achieving future negative emissions requires technologies (here excluding LULUCF) yet to be developed and that do not provide the important co-benefits of positive emissions reductions (e.g., energy security, health co-benefits). Since the responsibility-driven allocation of positive emissions ensures a given total emissions budget for each country, the capability-driven allocation of negative emissions results in an increase of near-term net emissions allocations for richer countries, as it otherwise reduces their longer-term net allocations. Many of these richer countries would have negative emissions budgets in 2021 already under a pure equal cumulative per capita allocation of global net emissions. An alternative parameterization of this Approach 1 could be to use HDI instead of GDP to better reflect the development of countries and their potential needs for development, supported by a view of development that is not purely economic[70]. Then, a country with a higher HDI can be allocated a greater effort as a share of negative emissions. However, using HDI as a capacity indicator may then penalize good governance compared to using GDP. Comparing two countries with equal populations and equal GDPs, the country with the higher HDI will have greater effort to provide when using HDI as the capacity indicator rather than GDP.

Practically, the first step of this approach is to annually allocate to countries negative emissions (excluding LULUCF) of the global scenario proportionally to their respective GDP projection (and thus indirectly based on their populations). Unlike Fyson et al.[47], the current approach does not filter out countries below the global mean of GDP per capita. As a second step, the positive emissions of the global

scenario are then allocated to equalize per capita emissions over the considered period and reflect historical responsibility (in terms of emissions since 1950 or 1990). The budget is equal to the cumulation of equal per capita emissions over that period. This modeling accounts for historical responsibility dynamically as the sum over time of equal per capita shares of the global emissions. The resulting budget matches that of a theoretical situation where countries had equal per capita emissions. This dynamic modeling of historical responsibility differs from a more integrated modeling where total emissions over the period considered are proportional to the total cumulated population over the same period. This budget can be negative for countries that had high emissions levels. Note that past emissions are first discounted by 1.5% each year in the past to account for technological improvement[71]. Each country is then allocated at every point in time (2021 to 2100) a fraction of the positive part of the global scenario proportional to its remaining budget. As a result, the first year's allocation differs from current emissions, and it may require IMTOs and very rapid scaling up of mitigation efforts to reconcile actual emissions with allocations over the period to 2030.

For both approaches, we define the budget $B$ as:

$$B_{Global} = \sum_{t=t_1}^{t_n} E_{Global}(t) \tag{1}$$

$$B_{Country} = \sum_{t_0}^{t_n} E_{Global}(t) \cdot \frac{\sum_{t_0}^{t_n} P_{Country}(t)}{\sum_{t_0}^{t_n} P_{Global}(t)} - \sum_{t^o}^{t_1} E_{Country}(t). \tag{2}$$

where $E(t)$ is the emissions of a country over time, $t_0 = 1990 (or 1950), t_n = 2100,$ and $t_1 = 2021$. Note that $B_{Country}$ can be negative for high historical per capita emitting countries.

For approach 1, the first step is to allocate the negative emissions of the global emissions pathway $E_{GlobalNeg}$ to countries:

$$E_{CountryNeg}(t) = E_{GlobalNeg}(t) \cdot \frac{GDP_{countries}(t)}{\sum_{i=countries} GDP_i(t)}, \forall t > t_1 \tag{3}$$

The next step is to allocate the positive emissions of the global emissions pathway $E_{GlobalPos}$ to countries

$$B_{CountryPos} = B_{Country} - \sum_{t_1}^{t_n} E_{CountryNeg}(t) \tag{4}$$

$$E_{CountryPos}(t) = \frac{B_{CountryPos}}{B_{GlobalPos}} \cdot E_{GlobalPos}(t), \forall t > t_1 \tag{5}$$

Finally:

$$E_{Country}(t) = E_{CountryNeg}(t) + E_{CountryPos}(t), \forall t > t_1 \tag{6}$$

Note that

$$\Sigma_{i=countries} E_i(t) = E_{Global}(t), \forall t \tag{7}$$

and that $\frac{\sum_{t_0}^{t_n} E_{Country}(t)}{\sum_{t_0}^{t_n} P_{Global}(t)}$ is equal for all countries.

We could not model the addition of a transition period to Approach 1 where long-term and near-term allocations are calculated jointly. Results of Approach 1 can be compared to the continuous modeling of the Equal Cumulative Per Capita allocation of ref. 3 to review the effect of a continuity criterion since both methods pursue equal historical responsibility by 2100.

The burden-sharing Approach 1 of the present study was used in the Traffic Light Reports[70] published by the Climate Vulnerable Forum (CVF) with specific parameterizations based on Gross Domestic Product (GDP) and Human Development Index (HDI) (Supplementary Discussion).

**Approach 2** models a capability-driven allocation of the global positive emissions and a responsibility-driven allocation of negative emissions. This modeling can reflect a funding of the near-term transition mostly by countries with the most financial capacities. Following these results, allocations can help mobilize the high investments needed to implement the mitigation measures across countries. Practically, the allocation of the positive part of the global scenario follows the approach of prior studies[3,54], where each country gets a share of global emissions proportional to its population divided by its GDP per capita dynamically (that is, at every point in time). This approach yields significant variations in emissions allocations across countries, which reflects the large differences across countries' GDPs (often proportionally greater than the differences in their historical contributions). This results in the allocation of important mitigation efforts for richer countries as a share of a global mitigation effort, which remains minor compared to global GDP[72].

As a second step, the allocation of the global negative emissions is proportional to countries' respective contributions to cumulative emissions dynamically (since 1950 or 1990 consistently with Approach 1 parameterization). Countries are allocated a share of the effort to contribute to removing emissions proportionally to their contribution to global warming at every point in time. This contribution to negative emissions is thus linked with their past population through their emissions but not linked to their future population. The influence of the responsibility component is entirely bounded by the levels of global negative emissions that grow over time. The starting point of the emissions allocation is thus hardly influenced by the responsibility component. The capability allocation contributes to reducing the difference in historical emissions across countries given the frequent correlation between countries' responsibility and capability.

The emissions allocations in the near term are driven by the GDP per capita of each country, which yields very large allocations for poor countries. For such countries, climate finance informed by Approach 2 allocations in this paper could increase their GDP indicators and thus reduce their allocation in turn. In practice, many of the poorest countries have committed to unconditional targets much lower than the equity-based allocations.

$$E_{CountryPos}(t) = E_{GlobalPos}(t) \cdot \frac{\frac{P_{Country}(t)^2}{GDP_{Country}(t)}}{\sum_{i=countries} P_i \left(\frac{t)^2}{GDP_{i}(t)}\right)}, \forall t > t_1. \tag{8}$$

$$B_{CountryNeg} = B_{Country} + \sum_{t_1}^{t_n} E_{CountryPos}(t) \tag{9}$$

$$E_{CountryNeg}(t) = \frac{B_{CountryNeg}}{B_{GlobalNeg}} \cdot E_{GlobalNeg}(t), \forall t > t_1. \tag{10}$$

### Ambition alignment of global emissions scenarios

Here, the warming alignment of a country's pledge reflects the global warming resulting from the emissions of the global scenarios whose allocation to that country is matched by its pledge. We use the representative scenarios from the IPCC-AR6 called C-scenarios, with warming outcomes ranging from 1.5 °C to above 4 °C. The respective emissions levels, including their negative emissions components, are not necessarily ordered according to their warming outcomes given

their underlying socio-economic assumptions. As a result, their respective emissions trajectories cross over time, which brings limitations in assessing the ambition of the NDCs of a small number of countries (see Supplementary Discussion).

The reference to a 1.5 °C alignment corresponds here to an alignment with the distribution of emissions of the average of scenarios of the IPCC Categories C1 (below 1.5 °C with no or limited overshoot). The distribution of C2 conveys a warming below 1.5 °C with high overshoot. The upper threshold of 2 °C alignment here follows the definition based on emissions scenarios C3 (likely below 2 °C) category. The consistency of such low emissions scenarios with the Paris Agreement temperature goal is discussed based on warming responses and levels of negative emissions[73,74]. Additionally, the effort-sharing formulas are applied to the scenario categories C6 (below 3 °C), C7 (below 4 °C) that reflect current policies, and C8 aligned with a warming above 4 °C. The allocation approaches can be applied to any global emissions scenarios, coherently with its socio-economic modeling assumptions. When averaging over multiple scenarios for each C-category, this study only considers the emissions of global scenarios but does not represent underlying socio-economic assumptions[75]. Avoiding any 1.5 °C overshoot and ensuring a higher likelihood of achieving that warming threshold thereby implies smaller emissions allocations still than the ones presented in this article.

The alignment of an NDC with a given emissions scenario is based on the unconditional part of the NDC, as it represents the mitigation effort provided by the country. When the emissions quantification of the NDC was provided with an uncertainty range, the alignment with a pathway is based on the average of the high and low values.

The absence of monotony between the warming response of the C-scenarios and their negative emissions can also result in non-monotonous 2030 allocations for some countries under Approach 1. In other words, some countries may have less stringent allocations under a 3 °C scenario than under 2 °C in 2030. While this paradox is compensated over time, such warming assessments are not relevant for these countries (list of countries in Supplementary Discussion).

Some of the selected global 1.5 °C scenarios with strong near-term mitigation also have positive emissions throughout the century when LULUCF emissions are excluded. As a result, some countries with relatively low responsibility and capability have emissions allocations that are positive throughout the century under a 1.5 °C objective.

## Data sources
The global emissions scenarios whose emissions are allocated to countries are the average of ensembles of scenarios in the categories C1 to C8 from the IPCC AR6 database[54] (accessible here). The GDP data (in purchasing power parity, ppp) is taken from the Social Socio-economic Pathways[55] associated with the global emissions scenarios (available here), specifically assuming the SSP2 scenario, describing a middle of the road between adaptation and mitigation challenges. Taking GDP without purchasing power parity correction could widen the difference in allocations between rich and poor countries. Historical emissions data is from a recent study[76] based on the Potsdam Real-time Integrated Model for the probabilistic Assessment of emission Paths (PRIMAP)[77,78] and the Global Carbon Project[76,79]. The population data is from the UN population prospects 2022 (available here). Bunkers scenario projections used in this article extrapolated from the ELEVATE project based on the IMAGE 3.4 model (available here, and visible in Supplementary Figs. 1 and 2). The quantification of NDCs is taken from a recent publication[80,81] (updated in March 2023). The country-level results are contingent on the limitations of the methods discussed above and on the limitation of the data projections used here. Population and especially GDP projections have intrinsic uncertainty that varies from country to country. In particular, GDP

projections for small countries should be seen as best guesses, and the resulting emissions allocations are indicative. Considering groups of small countries, possibly as their negotiating groups, can reduce the sensitivity of their emissions allocation to underlying data uncertainty. Additionally, the accounting of LULUCF emissions, here excluded, in reported data and emissions projections towards NDC targets can bring high uncertainty for countries with important forest coverage. Here, the data is coming from single sources for all countries, while the accounting of LULUCF in NDCs may differ from country to country and is often vaguely defined[58]. The allocation methods described here could be applied using other data projections, including governmental ones.

The countries considered are the 198 Paris to the UNFCCC. Parties for which data is missing are summarized in the Supplementary Tables 1 and 2 and discussed in the Supplementary Methods. Emissions allocations are run amongst countries with available data, and the emissions allocation of the EU is the sum of the allocations of its member states. Its allocation as a single entity would yield different results given the non-linearity of the effort-sharing formulae derived here. The same considerations apply to country groups.

## Reporting summary
Further information on research design is available in the Nature Portfolio Reporting Summary linked to this article.

## Data availability
All the material derived for the submission, including Supplementary Data with historical emissions accounted for since 1950 and 1990, is accessible for all countries online under the (https://doi.org/10.5281/zenodo.8003392). Data based on Approach 1 derived in the first submission was also used for the (https://fairsharenow.org) website and its Traffic Light Reports 2022 and 2024.

## Code availability
All code for computation, analysis and plotting is available via Zenodo at: (https://doi.org/10.5281/zenodo.13640303) and on Github at :(https://github.com/imagepbl/effort-sharing).

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

## Acknowledgements

This study was co-funded by Aroha and the Climate Vulnerable Forum (CVF). Details of the CVF's request are provided in the supplementary information. Y.Rd.P. is funded by the European Union's Horizon 2020 research and innovation program under the Marie Skłodowska-Curie grant agreement MSCA, project 101067708. The contribution of Dv.V. and M.D. was supported by funding from the European Union's Horizon Europe Research and Innovation Program under grant agreement No 101056873 (ELEVATE) and 101183367 (NewPathways).

## Author contributions

Y.R.d.P., M.D., and Dv.V. conceived the study. Y.R.d.P. designed the equity formulas. M.D. coded the implementation of the equity approaches and provided the figures. Y.R.d.P., M.D., M.S. and D.v.V. contributed to the data review and wrote the manuscript.

## Funding

## Competing interests

The authors declare no competing interests.
