## [Transparent Peer Review file · Nature Communications]

Effect of discontinuous fair-share emissions allocations immediately based on equity

Corresponding Author: Dr Yann Robiou du Pont

Version 0:

Reviewer comments:

Reviewer #1

(Remarks to the Author)

This paper presents a welcome methodological development to the analysis of NDC based on fairness/equity principles. The avoidance of using transition periods from current/baseline levels to 'fair' allocations has considerable implications for the distribution fair mitigation and finance efforts. I find Figure 4 particularly illuminating.

The presentation of the analysis and the language needs considerable improvement. This drawback also makes it difficult to fully assess the underlying method, although it appears to be sound. I would recommend to have the whole text copy-edited, in addition to addressing the specific issues listed below.

I would also like to see some more discussion of the merits and drawbacks of the paper's methodological contribution.

Most important points

1. The two fairness approaches are difficult to understand. The section in the main text should be made more accessible and related clearly to fairness principles in the literature. The differences between the two approaches should be highlighted. Why are the two approaches opposite wrt positive vs negative emissions?

143: [approach 1) "allocates global positive emissions to equalize historical responsibilities" . It is confusing that historical responsibility enters into approach 1, as I thought approach 1 was based on capability and approach 2 on responsibility. Also, historical responsibility over what time period?

147: "the capability considerations affects the use the emissions budgets over time, but not its total" Multiple typos here, and I do not understand the sentence. Affects the distribution over time, but not the total budget, maybe?

489: "This approach yields significant differences in emissions allocations across countries" Differences relative to what? Variation across countries might be a better term.

491-492: I do not understand this sentence. "Important" seems not the correct word here.

2. I would suggest to spend some more words on the criticism past studies have received for including transition periods (e.g., Kartha et al (2018)) and the weak ethical basis for such periods (e.g., Flerbaey et al (2014)). On the other hand, possible drawbacks of removing transition periods could also be discussed. For example, this means that reduced and avoided emissions are treated symmetrically, but the costs of reducing emissions are likely larger than the costs of avoiding future increases. The argument in the paper and in the literature against transition periods relies on emissions trading (ITMOs), but the imperfections of current institutions for this should be mentioned. Perhaps also refer to Knight (2013) for a defense of moderate grandfathering.

Sentences that are difficult to understand and should be reformulated:

17: "increasingly do so in the future" I could not understand what was meant until reading the introduction. This is an important point, but it needs to be elaborated more to be understandable in the abstract.

45: “most of the recent approaches rely on allocations of emissions rights following a continuous trajectory starting at current emissions levels, sometimes using a transition period” Why the word sometimes? Do not all continuous trajectories starting at current levels use a transition period by necessity? Or are there exceptions?. See also line 69.

131: “The relevance of equity concepts and their implementations in effort-sharing formulae show various consistency with international law”

167: “Approach 1 is mainly driven by responsibility in the near term” Change to (if I understand correctly: In the near term, a is driven mainly by responsibility

189: “The absence of zero or negative allocations for some countries results from fairness indicators as well as the absence of negative emissions in the 1.5°C scenarios-set with strong near-term mitigation, excluding LULUCF emissions.”

300: “The egalitarian approach...” Why is this introduced here? Difficult to follow.

343: “with few arbitrary” parameters. As few as possible? Or a few?

359: “Even equity-based budgets could theoretically be used mostly in the near-term by countries and not collectively reflect any of the global 1.5°C mitigation scenarios underpinning the global budget.”

364: “Additionally, emissions budgets are not suitable for addressing the knowledge gaps identified in the IPCC AR6 of “extending equity frameworks to quantify equitable international support, as the difference between equity-based national emissions scenarios and national domestic emissions scenarios””

457: “that may result from better governance or potentially ill acquired wealth”

524: “reflect alignment with symbolic warming thresholds”

527: “can be considered dragging even the insufficient ambition current policies that do not track towards NDCs”

Typos:

159: ‘nations’ should be notions?

246 “delay climate action and near 2030”

412: “that country could”

Minor points:

251: “Based on Approach 2, the assessment of the NDCs of the UK, Sweden and Switzerland is more stringent” Not apparent in the figure. Perhaps because they are nevertheless given the same assessment category?

Figure 2: For which countries is approach 2 less stringent than approach 1? I cannot see any country with a greener color under this approach, but perhaps the African countries are overdelivering 1.5degree ambition even more under approach 2?

387:” Compared to a previous warming assessment⁵ (visible on Paris-Equity-Check.org), Approach 1 finds NDCs to be more ambitious (1.5°C aligned) for a few countries (including India, Indonesia and Egypt) and less for Norway.” Are these these the only changes? Using what equity principle?

Request from the CVF: This section does not clearly distinguish between the request itself and the authors’ interpretation of it. The whole section is in quotation marks, but refers to “the CF requested” and “this paper”, which causes confusion over who formulated the text. I suggest replicating or summarizing the request first, then presenting the interpretation.

References

Fleurbaey, M., Kartha, S., Bolwig, S., Chee, Y.L., Chen, Y., Corbera, E., et al., 2014. Sustainable development and equity. In: Edenhofer, O., Pichs-Madruga, R., Sokona, Y., Farahani, E., Kadner, S., Seyboth, K. (Eds.), *Climate Change 2014: Mitigation of Climate Change. Contribution of Working Group III to the Fifth Assessment Report of the Intergovernmental Panel on Climate Change*. Cambridge University Press, Cambridge, United Kingdom and New York, NY, pp. 283–350.

Kartha, S., Athanasiou, T., Caney, S., Cripps, E., Dooley, K., Dubash, N.K., Harris, P., Holz, C., Lahn, B., Moellendorf, D., Müller, B., Roberts, J.T., Sagar, A.D., Shue, H., Singer, P., Winkler, H., 2018. Cascading biases against poorer parties. *Nat. Clim. Change* 8 (5), 348–349.

Knight, C., 2013. What is grandfathering? *Env. Polit.* 22 (3), 410–427.

Reviewer #2

(Remarks to the Author)

Robiou du Pont and co-authors note that most existing equitable mitigation assessments start from the most recent year when emissions are available for countries (termed "continuous allocations" by the authors). The authors indicate that this choice induces a "grandfathering" effect, which unintentionally rewards countries that have not reduced emissions with relatively less stringent emission reduction benchmarks in the near term. To address this, the authors propose an approach to derive equitable allocations for countries, which the authors suggest departs from previous literature in two ways: (1) the allocations start in a historical year (e.g., 1990), and hence capture historical (in-)action, and (2) the authors treat gross emission reductions and gross emission removals separately.

I have some comments and suggestions that I think are important to consider. I have focussed my review comments on the substantive content of the paper. However, I encourage the authors to try to reduce repetition between the "Introduction" and "Approach Rationale" sections, and to check for and correct typos and other errors in the text.

Review comments

Accounting for past (in-)action: I agree with the authors that updating an equitable mitigation assessment with updated historical emissions, all else equal, may result in an inadvertent benefit to emitters that are not reducing emissions (the argument the authors start to present in L44-L48). However, there are two things I think the authors should consider addressing:

1. The concept of "carbon debt": I think the same issue has been identified previously in the literature, where it has sometimes been termed as "carbon debt" or, emissions above a counterfactual equitable pathway (Gignac & Matthews, 2015; van den Berg et al., 2020). This approach does not fit neatly within the "continuous" versus "discontinuous" dichotomy that the authors have introduced.
2. The potential scale of the "grandfathering effect": The authors suggest, in L83-L85, that "Such iterative updates of ambition assessments based on continuous emissions allocations would iteratively find an insufficient NDC closer and closer to a calculated fair allocation". As I noted in the introduction to this section, I would tend to agree with this, all else equal. However, since pathways to a given temperature target (e.g., 1.5°C) will become progressively steeper, the responsibility of major emitters (which could be one equity allocation consideration) will increase. Given these two additional effects, I think there is more ambiguity in the effect on an equity assessment of NDCs. I suggest that the authors present some illustrative calculations to help the reader understand the validity of this statement.

Justification for the equity approaches: I appreciate that the authors provide a first estimation of the equitable mitigation targets that apply to both gross emission reductions as well as removals. As the authors correctly note, this is a gap in the existing literature, and addressing it is important to guide policy discussions. However, I have a few questions and concerns that I hope the authors can address:

1. I think the underlying justification for an equity approach is just as important as the numerical quantifications. Keeping this in mind, I found the justification for the application of different equity principles to gross reductions and removals to be one of the less comprehensive parts of the manuscript. I didn't understand why it is appropriate to factor in historical responsibility for one quantity (e.g., gross reductions) while factoring in capability for the other (e.g., gross removals). I think the manuscript could benefit from a more comprehensive discussion of the reasoning behind these equity approaches.
2. I found the description of the methods (L140 – L152 in the main text, and L436-L509 in the methods section) quite difficult to follow. I encourage the authors to publish the code used to carry out the analysis to allow for replication and to consider writing out the equations for each step so that the reader can follow the specific implementation.

Clarity on the use of scenarios: I have several comments on the use of global mitigation scenarios that I think the authors should address:

1. In L520-L522, the authors indicate that "The reference to a 1.5°C alignment corresponds to an alignment with the distribution of emissions of the average of scenarios of the IPCC Categories C1 [...], itself averaged with the distribution of C2 [...]". How do the authors come up with a distribution of emissions if they average across scenarios belonging to these categories? I'm not sure it is appropriate to use averages of such a scenario ensemble (see, e.g., (Guivarch et al., 2022)), and would recommend that the authors avoid this.
2. I think the authors should justify why they group the C1 and C2 categories of pathways together but do not do this for any of the other categories of pathways. Mapping the textual elements of the Paris Agreement Long Term Temperature Goal (LTTG) to specific pathway characteristics is a non-trivial value judgment (see, e.g., the discussions in (Kikstra et al., 2022; Schleussner et al., 2022)). I suggest that the authors improve the discussion on their pathway categorisation choices, especially since they explicitly indicate that this work is meant to guide the Global Stocktake.
3. Figure 2: The authors deviate from the labels presented in the methods section, by labeling C1+C2 pathways as "Below 1.5 degrees", and C3 pathways as "Well below 2 degrees". Please align the labels across the sections, and reflect on my comment above.
4. The 43% reduction by 2030 assessment: In L207-L209, the authors suggest that "The IPCC indicates that, on average across a set of scenarios, a 43% reduction in global GHG emissions by 2030 (here taken below 2020 levels) would align with a 1.5°C trajectory with no or limited overshoot. This global target [...]". There are a couple of conceptual challenges here, that I think the authors should consider addressing. The first, is that this is, by no means an IPCC-endorsed "global target" – it is only a description of the median (not average) of the scenarios assessed by the IPCC in that category of pathways. This value is also computed relative to 2019 emission levels, and given the structural differences between 2019 and 2020 emissions, I think it is further not appropriate to apply the 43% reduction below 2020 emission levels. Further, excluding LULUCF emissions (which are included in the original values presented in the IPCC report), means that this estimate is no longer appropriate. I suggest that the authors consider revising the text describing this approach, and use the uncertainty band presented in the AR6 report for this category while describing the results presented in Figure 3.

Additional analyses need to be motivated better: The two additional analyses presented in the results section (the addition of a “20-year transition phase”, and the presentation of equal per capita results) are not motivated sufficiently in the text. I was unsure why the authors chose to present these results and suggest the authors add more text before the results section to justify why they have presented them.

References

- Gignac, R., & Matthews, H. D. (2015). Allocating a 2 °C cumulative carbon budget to countries. *Environmental Research Letters*, 10(7), 075004. <https://doi.org/10.1088/1748-9326/10/7/075004>
- Guivarch, C., Le Gallic, T., Bauer, N., Fragkos, P., Huppmann, D., Jaxa-Rozen, M., Keppo, I., Kriegler, E., Krisztin, T., Marangoni, G., Pye, S., Riahi, K., Schaeffer, R., Tavoni, M., Trutnevyte, E., van Vuuren, D., & Wagner, F. (2022). Using large ensembles of climate change mitigation scenarios for robust insights. *Nature Climate Change*, 12(5), Article 5. <https://doi.org/10.1038/s41558-022-01349-x>
- Kikstra, J. S., Nicholls, Z. R., Smith, C. J., Lewis, J., Lamboll, R. D., Byers, E., Sandstad, M., Meinshausen, M., Gidden, M. J., Rogelj, J., & others. (2022). The IPCC Sixth Assessment Report WGIII climate assessment of mitigation pathways: From emissions to global temperatures. *Geoscientific Model Development*, 15(24), 9075–9109.
- Schleussner, C.-F., Ganti, G., Rogelj, J., & Gidden, M. J. (2022). An emission pathway classification reflecting the Paris Agreement climate objectives. *Communications Earth & Environment*, 3(1), <https://doi.org/10.1038/s43247-022-00467-w>
- van den Berg, N. J., van Soest, H. L., Hof, A. F., den Elzen, M. G. J., van Vuuren, D. P., Chen, W., Drouet, L., Emmerling, J., Fujimori, S., Höhne, N., Köberle, A. C., McCollum, D., Schaeffer, R., Shekhar, S., Vishwanathan, S. S., Vrontisi, Z., & Blok, K. (2020). Implications of various effort-sharing approaches for national carbon budgets and emission pathways. *Climatic Change*, 162(4), 1805–1822. <https://doi.org/10.1007/s10584-019-02368-y>

Version 1:

Reviewer comments:

Reviewer #1

(Remarks to the Author)

In the previous round, I noted that the presentation of the analysis and the language needed considerable improvement, and also recommended to have the whole text copy-edited. I regret to say that the improvement on these issues has not been satisfactory. It does not appear as the text has been copy-edited. It should not be the job of the reviewers to point out simple mistakes and help improve the language. While the authors have corrected the specific mistakes I pointed out, there are still many language problems, including in the newly introduced text. Some appear as sloppy mistakes, such as incomplete sentences, while a more extensive problem is lack of clear and structured presentation. I believe the underlying model development would be a valuable addition to the literature, but I deem the progress on its presentation from the first version as insufficient to warrant another ‘revise and resubmit’.

Some concrete issues:

Re. my first point that the two fairness approaches are difficult to understand, which was also brought up by R2: The motivation for including two different approaches is still not clear to me. The new text is quite technical. Do the approaches reflect different ethical assumptions, or different approaches to a more technical modeling choice for which there is no clear criterion for choosing one over the other?

The figures and captions appear in a mess. The same figures appear on multiple pages, sometimes with and sometimes without captions.

The discussion has no structure, which makes it difficult to follow. There is no conclusion.

The methods section appears to contain considerable overlap with the main text (partly reflecting that the main text is very technical).

Re. my suggestion to “spend some more words on the criticism past studies have received for including transition periods (e.g., Kartha et al (2018)) and the weak ethical basis for such periods (e.g., Fleurbaey et al (2014)). On the other hand, possible drawbacks of removing transition periods could also be discussed. For example, this means that reduced and avoided emissions are treated symmetrically, but the costs of reducing emissions are likely larger than the costs of avoiding future increases. The argument in the paper and in the literature against transition periods relies on emissions trading (ITMOs), but the imperfections of current institutions for this should be mentioned. Perhaps also refer to Knight (2013) for a defense of moderate grandfathering.”

The authors responded ‘Thank you for the reference. We added it in the following sentence as follows with references to Fleurbaey and Knight:

“Considering continuous emissions trajectories that look realistic²² implies that present-day levels of domestic emissions are an acceptable starting point in terms of mitigation effort with a utilitarian perspective²⁵.” ‘

I would have liked to see a more engagement with the suggestion than just adding one sentence.

(Remarks on code availability)

Reviewer #2

(Remarks to the Author)

Thank you for the opportunity to review the revised manuscript.

I have reviewed the revisions made by the authors in response to my previous round of review comments. I am satisfied with these revisions.

(Remarks on code availability)

Version 2:

Reviewer comments:

Reviewer #1

(Remarks to the Author)

(Remarks on code availability)

REVIEWER COMMENTS

Reviewer #1 (Remarks to the Author):

This paper presents a welcome methodological development to the analysis of NDC based on fairness/equity principles. The avoidance of using transition periods from current/baseline levels to ‘fair’ allocations has considerable implications for the distribution fair mitigation and finance efforts. I find Figure 4 particularly illuminating.

The presentation of the analysis and the language needs considerable improvement. This drawback also makes it difficult to fully assess the underlying method, although it appears to be sound. I would recommend to have the whole text copy-edited, in addition to addressing the specific issues listed below.

I would also like to see some more discussion of the merits and drawbacks of the paper’s methodological contribution.

Most important points

1. The two fairness approaches are difficult to understand. The section in the main text should be made more accessible and related clearly to fairness principles in the literature. The differences between the two approaches should be highlighted.

Why are the two approaches opposite wrt positive vs negative emissions?

Thank you. We extended the discussion to address these shortcomings:

“Here, we suggest two extensions, one for each approach of Fyson et al. to derive two allocations of economy-wide emissions to countries. Each new approach combines concepts of capability and responsibility, where each concept is applied to global positive or negative emissions distinctively. This study offers two conceptual combinations of the responsibility and capability concepts referred to in the Paris Agreement’s CBDR-RC, with a differentiated treatment of negative emissions, which require costly and uncertain technologies, and are made necessary because of insufficient global emissions reductions to date. These approaches enable the assessment of the ambition of countries’ NDC, in light of their responsibility and capability, with special considerations for negative emissions often used to enable and justify potentially dangerous warming overshoot. A first extension, named Approach 1, first allocates global negative emissions across countries based on their capability, assessed through GDP or Human Development Index (HDI, in Supplementary Information), and then allocates global positive emissions to equalize historical responsibilities over the net emissions (positive + negative, see Methods). Under this approach, rich countries are required to fund most of the negative emissions that require important research and development costs with high uncertainty and without local co-benefits³¹. Approach 1 also ensures equal cumulative per capita emissions by 2100 through the allocation of the positive emissions space. Here, the capability allocation affects the distribution of emissions over time, but not the total budget. Richer countries then have more important negative allocations in the future, and less stringent allocations in the near term (see Methods). The second extension, Approach 2, conversely first allocates global positive emissions based on countries’ capabilities and then global negative emissions based on their responsibilities. There, all countries contribute to emissions reductions based on their wealth. Negative emissions, needed because of the world’s important historical

emissions, are allocated proportionally to countries' individual historical responsibilities. In Approach 2, historical responsibility does not define countries' cumulative emissions alone. Looking at the global emissions scenarios, the positive emissions refer here to the projected physical emissions (e.g., fossil fuels, agriculture). The negative emissions here refer to emissions captured through Carbon Dioxide Removal (excluding those from LULUCF, unlike Fyson et al. 2020) and Direct Air Capture.

143: [approach 1) "allocates global positive emissions to equalize historical responsibilities" . It is confusing that historical responsibility enters into approach 1, as I thought approach 1 was based on capability and approach 2 on responsibility. Also, historical responsibility over what time period?

Indeed, the text was misleading. We added a sentence explaining the rationale of the paper that is to suggest two extensions of the Fyson et al. paper, each of which combines responsibility (since 1990 and 1950) and capability. Extending each of the two approaches of Fyson et al. For clarity, we amended the previous paragraph and added the sentence line 143:

"Each new approach combines concepts of capability and responsibility, where each concept is applied to global positive or negative emissions distinctively."

And in the following paragraph on the parameterization:

"Here, we present results based on a parameterization that uses GDP for capability and accounts for responsibility through emissions since 1990 (with 1950 in the Supplementary Information)."

147: "the capability considerations affects the use the emissions budgets over time, but not its total" Multiple typos here, and I do not understand the sentence. Affects the distribution over time, but not the total budget, maybe?

Thanks for pointing this out and the concrete suggestion. The sentence was modified according to the suggestion:

"Approach 1 also ensures equal cumulative per capita emissions by 2100 through the allocation of the positive emissions space. Here, the capability allocation affects the distribution of emissions over time, but not the total budget. Richer countries then have more important negative allocations in the future, and less stringent allocations in the near term (see Methods)."

489: "This approach yields significant differences in emissions allocations across countries" Differences relative to what? Variation across countries might be a better term.

Thank you. I had not realized the difference in meanings of these two terms. The text was modified according to this suggestion.

491-492: I do not understand this sentence. "Important" seems not the correct word here.

Thank you. Would this phrasing be clearer?

“This approach yields significant variations in emissions allocations across countries, which reflects the large differences across countries’ GDPs (often proportionally greater than the differences of their historical contributions).”

2. I would suggest to spend some more words on the criticism past studies have received for including transition periods (e.g., Kartha et al (2018)) and the weak ethical basis for such periods (e.g., Fleurbaey et al (2014)). On the other hand, possible drawbacks of removing transition periods could also be discussed. For example, this means that reduced and avoided emissions are treated symmetrically, but the costs of reducing emissions are likely larger than the costs of avoiding future increases. The argument in the paper and in the literature against transition periods relies on emissions trading (ITMOs), but the imperfections of current institutions for this should be mentioned. Perhaps also refer to Knight (2013) for a defense of moderate grandfathering.

Thank you for the reference. We added it in the following sentence as follows with references to Fleurbaey and Knight:

“Considering continuous emissions trajectories that look realistic²² implies that present-day levels of domestic emissions are an acceptable starting point in terms of mitigation effort with a utilitarian perspective²⁵.”

Sentences that are difficult to understand and should be reformulated:

17: “increasingly do so in the future” I could not understand what was meant until reading the introduction. This is an important point, but it needs to be elaborated more to be understandable in the abstract.

Certainly. We reformulate to:

“Ambition assessments based on trajectories that start at present-day emissions levels inherently reward past inaction, and increasingly do so with their iterative updates.”

45: “most of the recent approaches rely on allocations of emissions rights following a continuous trajectory starting at current emissions levels, sometimes using a transition period” Why the word sometimes? Do not all continuous trajectories starting at current levels use a transition period by necessity? Or are there exceptions?. See also line 69.

Thanks for raising this. Indeed, the continuity of emissions allocations does not require a transition period. The formula can be designed in other ways. To clarify this point, we suggest changing the text shortly after (originally on line 73):

“Effort-sharing formulas can be designed to directly achieve such continuity^{2,3} or an ad-hoc transition period can be added to ensure continuity^{3,14,16} between current emissions and future allocations only based on fairness considerations.”

This is also further explained in the discussion section:

“In other models, the continuity of the emissions trajectory results from the choice of allocating mitigation burden to depart from a reference trajectory rather than allocating the remaining emissions space². Such an approach based on reference trajectories can be adequate to assess the ambition of an emissions target when provided with a corresponding reference scenario, e.g. pledges taken in 2015 against allocations starting in 2015 (ref. 2).”

131: “The relevance of equity concepts and their implementations in effort-sharing formulae show various consistency with international law”

Thank you. We suggested this formulation:

“Some of the equity concepts quantified in the literature are not backed by principles of international law that require excluding approaches based on grandfathering⁶.”

167: “Approach 1 is mainly driven by responsibility in the near term” Change to (if I understand correctly: In the near term, a1 is driven mainly by responsibility

Thank you for the concrete suggestion. We decided to delete this sentence.

189: “The absence of zero or negative allocations for some countries results from fairness indicators as well as the absence of negative emissions in the 1.5°C scenarios-set with strong near-term mitigation, excluding LULUCF emissions.”

We suggest reformulating to two sentences:

“Looking at when allocations reach net-zero, some countries with relatively low responsibility and capability have emissions allocations that are positive throughout the century under a 1.5°C objective. It is important to note that some of the selected global 1.5°C scenarios with strong near-term mitigation also have positive emissions throughout the century since we excluded the LULUCF sector.”

300: “The egalitarian approach...” Why is this introduced here? Difficult to follow.

Thanks for highlighting this. We added an introductory sentence and modified the previous one to:

“In addition to capability and responsibility, equality is the third equity principle described in the IPCC AR5^{17,22}. IPCC reports do not present equity-based emissions allocations since AR5, despite available studies and its importance for courts of law¹². The egalitarian approach modelled as equal per capita emissions is not directly anchored in the Paris Agreement or international environmental law⁶, but can reveal the inequalities of emissions spaces claimed through NDCs.”

343: “with few arbitrary” parameters. As few as possible? Or a few?

We changed wording to: “as few parameters as possible”

359: “Even equity-based budgets could theoretically be used mostly in the near-term by countries and not collectively reflect any of the global 1.5°C mitigation scenarios underpinning the global budget.”

We suggest simplifying the sentence to:

“Theoretically, countries could choose to use a budget mostly in the near-term to justify insufficient emissions objectives, which raises issues of intergenerational justice.”

364: “Additionally, emissions budgets are not suitable for addressing the knowledge gaps identified in the IPCC AR6 of “extending equity frameworks to quantify equitable international support, as the difference between equity-based national emissions scenarios and national domestic emissions scenarios””

In response, we modified the previous few sentences for clarity and amended these two sentences to:

“Theoretically, countries could choose to use a budget mostly in the near-term to justify insufficient emissions objectives, which raises issues of intergenerational justice. The “flexibility” provided by carbon budgets over emissions pathways¹⁴ comes at the expense of the ability to assess the ambition of time-defined objectives, without additional assumptions¹⁸. Time-defined emissions allocations are also needed to address the knowledge gaps identified in the IPCC AR6 of “extending equity frameworks to quantify equitable international support, as the difference between equity-based national emissions scenarios and national domestic emissions scenarios”⁵².”

457: “that may result from better governance or potentially ill acquired wealth”

We suggest reformulating to:

“Comparing two countries with equal populations and equal GDPs, the country with higher HDI will have greater effort to provide when using HDI as the capacity indicator rather than GDP. Using HDI as a capacity indicator may then penalize good governance, compared to using GDP. Results based on HDI are available in the supplementary information.”

524: “reflect alignment with symbolic warming thresholds”

We simplified the phrasing to:

“Additionally, the effort-sharing formulas are applied to the scenario categories C6 (‘below 3°C’) and C7 (below 4°C) that reflect current policies.”

527: “can be considered dragging even the insufficient ambition current policies that do not track towards NDCs”

We simplified to:

“Countries with NDCs that do not align with their fair allocation of C7 scenarios can be considered to be dragging global decarbonization efforts.”

Typos:

159: ‘nations’ should be notions?

Yes. This typo is corrected.

246 “delay climate action and near 2030”

We clarified the sentence to:

“This effect increases as we delay climate action and as we near 2030.”

412: “that country could”

Indeed, corrected to “countries”.

Minor points:

251: “Based on Approach 2, the assessment of the NDCs of the UK, Sweden and Switzerland is more stringent” Not apparent in the figure. Perhaps because they are nevertheless given the same assessment category?

Thank you. We have now corrected to:

“Based on Approach 2, the assessment of the NDCs of the UK and Switzerland is more stringent given their relatively low historical responsibility compared to their relatively high capability.”

Figure 2: For which countries is approach 2 less stringent than approach 1? I cannot see any country with a greener color under this approach, but perhaps the African countries are overdelivering 1.5degree ambition even more under approach 2?

Indeed, Figure 3 (formerly figure 2) provides a rating of NDC ambition. The full allocations are provided in the supplementary data that will be made available publicly for all countries (the version initially submitted is available here <https://zenodo.org/record/8003393>). We added a new sentence after the two sentences that discussed the relative stringencies of approaches 1 & 2:

“While Approach 1 constrains countries’ cumulative emissions based on their responsibility, which often overlaps with high capability³⁸, Approach 2 is more stringent in the near term for countries with high GDP. Approach 2 is less stringent for countries with very low capability (sub-Saharan African countries), and for countries with high historical responsibility (USA, Russia, Qatar and other fossil fuel extracting countries, Figure 2 and Supplementary Information). These different stringencies of emissions allocations do not always change the warming assessment of countries’ NDCs (see country-level allocations in SI).”

387:” Compared to a previous warming assessment⁵ (visible on Paris-Equity-Check.org), Approach 1 finds NDCs to be more ambitious (1.5°C aligned) for a few countries (including India, Indonesia and Egypt) and less for Norway.” Are these these the only changes? Using what equity principle?

This previous assessment (which I am also author of), had different methods combining different equity principles. Here, we choose to report the most prominent differences. We suggest clarifying that this previous assessment relied on three equity principles:

“Compared to a previous warming assessment⁵ (visible on Paris-Equity-Check.org) where each country follows the least stringent of three equity principles (capability, responsibility, and equality), the present approaches find NDCs to be more ambitious (1.5°C aligned) for a few countries (including India, Indonesia, and Egypt depending on the approach) and less for countries in the Global North and Latin America.”

Request from the CVF: This section does not clearly distinguish between the request itself and the authors’ interpretation of it. The whole section is in quotation marks, but refers to “the CF requested” and “this paper”, which causes confusion over who formulated the text. I suggest replicating or summarizing the request first, then presenting the interpretation.

Thank you. We seek to provide transparency with this section. Your input is helpful in that regard. We changed the titles of the two sections as follows:

Request as explained by the CVF:

Interpretation and discussion of CVF’s request in light of the available literature:

References

Fleurbaey, M., Kartha, S., Bolwig, S., Chee, Y.L., Chen, Y., Corbera, E., et al., 2014. Sustainable development and equity. In: Edenhofer, O., Pichs-Madruga, R., Sokona, Y., Farahani, E., Kadner, S., Seyboth, K. (Eds.), Climate Change 2014: Mitigation of Climate Change. Contribution of Working Group III to the Fifth Assessment Report of the

Intergovernmental Panel on Climate Change. Cambridge University Press, Cambridge, United Kingdom and New York, NY, pp. 283–350.

Kartha, S., Athanasiou, T., Caney, S., Cripps, E., Dooley, K., Dubash, N.K., Harris, P., Holz, C., Lahn, B., Moellendorf, D., Müller, B., Roberts, J.T., Sagar, A.D., Shue, H., Singer, P., Winkler, H., 2018. Cascading biases against poorer parties. *Nat. Clim. Change* 8 (5), 348–349.

Knight, C., 2013. What is grandfathering? *Env. Polit.* 22 (3), 410–427.

Reviewer #2 (Remarks to the Author):

Robiou du Pont and co-authors note that most existing equitable mitigation assessments start from the most recent year when emissions are available for countries (termed "continuous allocations" by the authors). The authors indicate that this choice induces a "grandfathering" effect, which unintentionally rewards countries that have not reduced emissions with relatively less stringent emission reduction benchmarks in the near term. To address this, the authors propose an approach to derive equitable allocations for countries, which the authors suggest departs from previous literature in two ways: (1) the allocations start in a historical year (e.g., 1990), and hence capture historical (in-)action, and (2) the authors treat gross emission reductions and gross emission removals separately.

I have some comments and suggestions that I think are important to consider. I have focussed my review comments on the substantive content of the paper

Review comments

Accounting for past (in-)action: I agree with the authors that updating an equitable mitigation assessment with updated historical emissions, all else equal, may result in an inadvertent benefit to emitters that are not reducing emissions (the argument the authors start to present in L44-L48). However, there are two things I think the authors should consider addressing: 1. The concept of "carbon debt": I think the same issue has been identified previously in the literature, where it has sometimes been termed as "carbon debt" or, emissions above a counterfactual equitable pathway (Gignac & Matthews, 2015; van den Berg et al., 2020). **This approach does not fit neatly within the "continuous" versus "discontinuous" dichotomy that the authors have introduced.**

I understand the carbon debt as related to the accounting of historical emissions. The disparities in historical emissions across countries can be seen as a carbon debt from the high historical emitters, including to the lower emitting countries. Previous studies, including (Gignac & Matthews, 2015; van den Berg et al., 2020), suggested that future emissions allocations can account for this debt. The debt could be compensated over the period for which the future emissions budget is allocated, or possibly through adaptation and loss-and-damage finance. In a recent submission, Pelz et al. (<http://dx.doi.org/10.21203/rs.3.rs-4394688/v1>, under review) highlight the importance of this debt to "(i) responsibility for overshoot, ii) exceedance drawdown obligations, and iii) increase in extreme climate exposure if drawdown does not occur." While our approach allocates immediate effort to

compensate inequities, the “carbon debt” tracks overshoot but does not suggest how this debt should be compensated over time.

Our present study and its use of discontinuous dynamic emissions allocations does not differ in that aspect. The carbon debt is accounted for and influences future emissions allocations, even resulting in equal cumulative per capita emissions in Approach 1. In this case, the ‘discontinuity’ feature only affects how the emissions budget is used over time.

We clarify this point in the text as:

“Other approaches use the concept of ‘carbon debt’^{55,56} to assess countries’ responsibility for overshooting their fair shares of a global carbon budget, including through their future objectives, and compensate through future negative emissions⁵⁷. This budget-based approach can be used to characterize breaches by courts and be complementary to dynamic approaches, as in this study, immediately allocating feasible socio-economic scenarios, which can inform Paris-aligned emissions targets.”

2. The potential scale of the “grandfathering effect”: The authors suggest, in L83-L85, that “Such iterative updates of ambition assessments based on continuous emissions allocations would iteratively find an insufficient NDC closer and closer to a calculated fair allocation”. As I noted in the introduction to this section, I would tend to agree with this, all else equal. However, since pathways to a given temperature target (e.g., 1.5°C) will become progressively steeper, the responsibility of major emitters (which could be one equity allocation consideration) will increase. Given these two additional effects, I think there is more ambiguity in the effect on an equity assessment of NDCs. **I suggest that the authors present some illustrative calculations to help the reader understand the validity of this statement.**

Thank you. We updated Figure 1 to use actual data instead of the schematic representation in the first submission. We modelled allocation starting in 2015 and 2020. As you guessed, the slope is steeper in the later allocation, but the grandfathering effect remains visible. Thank you for the suggestion.

Justification for the equity approaches: I appreciate that the authors provide a first estimation of the equitable mitigation targets that apply to both gross emission reductions as well as removals. As the authors correctly note, this is a gap in the existing literature, and addressing it is important to guide policy discussions. However, I have a few questions and concerns that I hope the authors can address:

1. I think the underlying justification for an equity approach is just as important as the numerical quantifications. Keeping this in mind, I found the justification for the application of different equity principles to gross reductions and removals to be one of the less comprehensive parts of the manuscript. I didn’t understand why it is appropriate to factor in historical responsibility for one quantity (e.g., gross reductions) while factoring in capability for the other (e.g., gross removals). I think the manuscript could benefit from a more comprehensive discussion of the reasoning behind these equity approaches.

Thank you, the lack of clarity of this section was also raised by reviewer 1. We agree with the importance of the justification updated the description as follows:

“Here, we suggest two extensions, one for each approach of Fyson et al. to derive two allocations of economy-wide emissions to countries. Each new approach combines concepts of capability and responsibility, where each concept is applied to global positive or negative emissions distinctively. This study offers two conceptual combinations of the responsibility and capability concepts referred to in the Paris Agreement’s CBDR-RC, with a differentiated treatment of negative emissions, which require costly and uncertain technologies, and are made necessary because of insufficient global emissions reductions to date. These approaches enable the assessment of the ambition of countries’ NDC, in light of their responsibility and capability, with special considerations for negative emissions often used to enable and justify potentially dangerous warming overshoot. A first extension, named Approach 1, first allocates global negative emissions across countries based on their capability, assessed through GDP or Human Development Index (HDI, in Supplementary Information), and then allocates global positive emissions to equalize historical responsibilities over the net emissions (positive + negative, see Methods). Under this approach, rich countries are required to fund most of the negative emissions that require important research and development costs with high uncertainty and without local co-benefits³¹. Approach 1 also ensures equal cumulative per capita emissions by 2100 through the allocation of the positive emissions space. Here, the capability allocation affects the distribution of emissions over time, but not the total budget. Richer countries then have more important negative allocations in the future, and less stringent allocations in the near term (see Methods). The second extension, Approach 2, conversely first allocates global positive emissions based on countries’ capabilities and then global negative emissions based on their responsibilities. There, all countries contribute to emissions reductions based on their wealth. Negative emissions, needed because of the world’s important historical emissions, are allocated proportionally to countries’ individual historical responsibilities. In Approach 2, historical responsibility does not define countries’ cumulative emissions alone. Looking at the global emissions scenarios, the positive emissions refer here to the projected physical emissions (e.g., fossil fuels, agriculture). The negative emissions here refer to emissions captured through Carbon Dioxide Removal (excluding those from LULUCF, unlike Fyson et al. 2020) and Direct Air Capture.”

2. I found the description of the methods (L140 – L152 in the main text, and L436-L509 in the methods section) quite difficult to follow. I encourage the authors to publish the code used to carry out the analysis to allow for replication and to consider writing out the equations for each step so that the reader can follow the specific implementation.

Thank you. We will make the code publicly available with the manuscript on a Github. It is shared with this version of the manuscript through Code Ocean and at: <https://github.com/imagepbl/effort-sharing>. We also added a description of the formulas used in the Methods.

Clarity on the use of scenarios: I have several comments on the use of global mitigation scenarios that I think the authors should address:

1. In L520-L522, the authors indicate that “The reference to a 1.5°C alignment corresponds to an alignment with the distribution of emissions of the average of scenarios of the IPCC

Categories C1 [...], itself averaged with the distribution of C2 [...].” How do the authors come up with a distribution of emissions if they average across scenarios belonging to these categories? I’m not sure it is appropriate to use averages of such a scenario ensemble (see, e.g., (Guivarch et al., 2022)), and would recommend that the authors avoid this.

Thank you for raising this. Guivarch et al. 2022 (co-authored by one of the authors of the present manuscript) highlight: “Although scenario ensembles are designed to explore the possibility space, neither type of ensemble can be interpreted as a perfect statistical sample. Given the unknown unknowns, the scenarios’ outcomes cannot be interpreted in terms of likelihoods, and even large scenario ensembles do not fully or equally explore the space of possibilities”.

We agree that averaging across the ensemble of scenarios in C-categories does not provide a representative view of the literature as some modeling choices may be over/under-represented. However, our analysis uses global scenarios only for their global emissions bound by common physical considerations. The socio-economic assumptions of these scenarios have limited effect on the total emissions profile, mostly driven by considerations regarding total negative emissions. This approach is consistent with previous studies, including:

Robiou du Pont, Y., Jeffery, M., Gütschow, J. et al. Equitable mitigation to achieve the Paris Agreement goals. *Nature Clim Change* 7, 38–43 (2017).

Xunzhang Pan, Michel den Elzen, Niklas Höhne, Fei Teng, Lining Wang, Exploring fair and ambitious mitigation contributions under the Paris Agreement goals, *Environmental Science & Policy*, Volume 74, 2017, Pages 49-56, ISSN 1462-9011, <https://doi.org/10.1016/j.envsci.2017.04.020>.

Note that part of the author team is working on another article, under review, suggesting custom-made global emissions scenario representatives of various warming categories, that enables the user to check the influence of modelling parameters separately:

Mark Dekker, Andries Hof, Yann Robiou du Pont et al. Navigating the black box of fair national emissions targets, 19 September 2024, PREPRINT (Version 1) available at Research Square [<https://doi.org/10.21203/rs.3.rs-5023350/v1>]

2. I think the authors should justify why they group the C1 and C2 categories of pathways together but do not do this for any of the other categories of pathways. Mapping the textual elements of the Paris Agreement Long Term Temperature Goal (LTTG) to specific pathway characteristics is a non-trivial value judgment (see, e.g., the discussions in (Kikstra et al., 2022; Schleussner et al., 2022)). I suggest that the authors improve the discussion on their pathway categorisation choices, especially since they explicitly indicate that this work is meant to guide the Global Stocktake.

Thank you. We understand the critics of Schleussner et al. regarding the potential misinterpretation of scenario categories, their absence of overlaps and their representativeness of the Paris Agreement goal. We now provide the results for C1 and C2 separately and amended the text in the following manner:

“The reference to a 1.5°C alignment corresponds here to an alignment with the

distribution of emissions of the average of scenarios of the IPCC Categories C1 ('below 1.5°C with no or limited overshoot'). The distribution of C2 conveys a warming 'below 1.5°C with high overshoot'. The upper threshold of 2°C alignment here follows the definition based on emissions scenarios C3 ('likely below 2°C') category. The consistency of such low emissions scenarios with the Paris Agreement temperature goal is discussed based on warming responses and levels of negative emissions^{62,63}.”

3. Figure 2: The authors deviate from the labels presented in the methods section, by labeling C1+C2 pathways as “Below 1.5 degrees”, and C3 pathways as “Well below 2 degrees”.
Please align the labels across the sections, and reflect on my comment above.

Thank you. We have corrected the labelling.

4. The 43% reduction by 2030 assessment: In L207-L209, the authors suggest that “The IPCC indicates that, on average across a set of scenarios, a 43% reduction in global GHG emissions by 2030 (here taken below 2020 levels) would align with a 1.5°C trajectory with no or limited overshoot. This global target [...]”. There are a couple of conceptual challenges here, that I think the authors should consider addressing. The first, is that this is, by no means an IPCC-endorsed “global target” – it is only a description of the median (not average) of the scenarios assessed by the IPCC in that category of pathways. This value is also computed relative to 2019 emission levels, and given the structural differences between 2019 and 2020 emissions, I think it is further not appropriate to apply the 43% reduction below 2020 emission levels. Further, excluding LULUCF emissions (which are included in the original values presented in the IPCC report), means that this estimate is no longer appropriate. I suggest that the authors consider revising the text describing this approach, and use the uncertainty band presented in the AR6 report for this category while describing the results presented in Figure 3.

Thank you. We updated the data to show a 50% reduction below 2020 levels and no longer refer to the IPCC figure to simply illustrate national allocation at a midway point of global decarbonization. Figure 2 (formerly Figure 3) shows this new parameterization. Thank you for the suggestion that avoids misinterpretation.

Additional analyses need to be motivated better: The two additional analyses presented in the results section (the addition of a “20-year transition phase”, and the presentation of equal per capita results) are not motivated sufficiently in the text. I was unsure why the authors chose to present these results and suggest the authors add more text before the results section to justify why they have presented them.

Thank you. We added the following explanations.

First we show equal per capita allocations as a point of comparison, revealing some lack of ambition even in the absence of equity considerations.

“In addition to capability and responsibility, equality is the third equity principle described in the IPCC AR5^{17,22}. IPCC reports do not present equity-based emissions allocations since AR5, despite available studies and its importance for courts of

law12. The egalitarian approach modelled as equal per capita emissions is not directly anchored in the Paris Agreement or international environmental law⁶, but can reveal the inequalities of emissions spaces claimed through NDCs.”

Secondly, we explain how modelling discontinuous emissions allocations changes not simply countries’ fair share, but also the ranking of countries in terms of the relative share of global mitigation finance that they could be expected to contribute to. It is not only important in terms of which country is doing enough or not, but also to determine the distribution of expected financial effort across countries. We modify the introductory text of Figure 4’s results to:

“We show that modelling continuous emissions allocations, here exemplified by adding a 20-year transition period, affects countries’ emissions gaps between their allocations and NDCs, unequally in 2030. Compared to a traditional continuous approach, applying a discontinuous approach implies here a much higher obligation to contribute to international finance for all G20 countries except from India. In terms of the ranking of the emissions gap between NDC and allocation, assuming a transition period benefits Canada and Australia (moving down 9 positions), the USA and South Korea (each 8 positions). This shows that continuous pathways ‘reward’ such countries for their history of comparably low mitigation efforts, lowering their implied contribution to international climate finance. Other countries, including China, Türkiye, South Africa and the EU move down in the ranking of the ambition gaps, when removing the transition period.”

References

- Gignac, R., & Matthews, H. D. (2015). Allocating a 2 °C cumulative carbon budget to countries. *Environmental Research Letters*, 10(7), 075004. <https://doi.org/10.1088/1748-9326/10/7/075004>
- Guivarch, C., Le Gallic, T., Bauer, N., Fragkos, P., Huppmann, D., Jaxa-Rozen, M., Keppo, I., Kriegler, E., Krisztin, T., Marangoni, G., Pye, S., Riahi, K., Schaeffer, R., Tavoni, M., Trutnevyte, E., van Vuuren, D., & Wagner, F. (2022). Using large ensembles of climate change mitigation scenarios for robust insights. *Nature Climate Change*, 12(5), Article 5. <https://doi.org/10.1038/s41558-022-01349-x>
- Kikstra, J. S., Nicholls, Z. R., Smith, C. J., Lewis, J., Lamboll, R. D., Byers, E., Sandstad, M., Meinshausen, M., Gidden, M. J., Rogelj, J., & others. (2022). The IPCC Sixth Assessment Report WGIII climate assessment of mitigation pathways: From emissions to global temperatures. *Geoscientific Model Development*, 15(24), 9075–9109.
- Schleussner, C.-F., Ganti, G., Rogelj, J., & Gidden, M. J. (2022). An emission pathway classification reflecting the Paris Agreement climate objectives. *Communications Earth & Environment*, 3(1), <https://doi.org/10.1038/s43247-022-00467-w>
- van den Berg, N. J., van Soest, H. L., Hof, A. F., den Elzen, M. G. J., van Vuuren, D. P., Chen, W., Drouet, L., Emmerling, J., Fujimori, S., Höhne, N., Köberle, A. C., McCollum, D., Schaeffer, R., Shekhar, S., Vishwanathan, S. S., Vrontisi, Z., & Blok, K. (2020). Implications of various effort-sharing approaches for national carbon budgets and emission pathways.

Climatic Change, 162(4), 1805–1822. <https://doi.org/10.1007/s10584-019-02368-y>

REVIEWER COMMENTS

Reviewer #1 (Remarks to the Author):

In the previous round, I noted that the presentation of the analysis and the language needed considerable improvement, and also recommended to have the whole text copy-edited. I regret to say that the improvement on these issues has not been satisfactory. It does not appear as the text has been copy-edited. It should not be the job of the reviewers to point out simple mistakes and help improve the language. While the authors have corrected the specific mistakes I pointed out, there are still many language problems, including in the newly introduced text. Some appear as sloppy mistakes, such as incomplete sentences, while a more extensive problem is lack of clear and structured presentation. I believe the underlying model development would be a valuable addition to the literature, but I deem the progress on its presentation from the first version as insufficient to warrant another 'revise and resubmit'.

We appreciate that improvements to the text were necessary and are thankful that you took the time to go over it one more time. We also apologise the display issues due to the conversion of the word document into a pdf file (in particular figure captions).

We now copy-edited the entire manuscript, reorganised sections for clarity, and removed redundant statements.

Some concrete issues:

Re. my first point that the two fairness approaches are difficult to understand, which was also brought up by R2: The motivation for including two different approaches is still not clear to me. The new text is quite technical. Do the approaches reflect different ethical assumptions, or different approaches to a more technical modeling choice for which there is no clear criterion for choosing one over the other?

Thank you. There are indeed several motivations behind these modelling choices.

- 1) combining capability and responsibility without relying on averages or statistical combinations,
- 2) differentiating the allocation of positive and negative emissions, to complement the study of Fyson et al. that focused on the allocation of global negative emissions only (with either responsibility or capability)

Given these choices, we chose simple allocation methods to represent the capability and the responsibility principles. We model the two manners to combine the responsibility and capability allocations of positive and negative emissions separately. We explain the differences (e.g., Approach 1 achieves equal cumulative per capita emissions) but do not suggest that one should be used over the other.

We have updated the manuscript to clarify the rationale of the methodology:

“Here we quantify two sets of emissions trajectories immediately based on equity principles and that do not start at current emissions levels (see Methods). The two

methods combine the equity principles of capability and responsibility¹ to reflect the principles of the UNFCCC and the Paris Agreement, notably CBDR-RC. The literature suggests several approaches, conceptual or statistical, for combining different equity principles into a single allocation method (see Discussion). Here we apply each of the two equity principles to allocate global positive or negative emissions separately. This differentiated treatment of negative emissions extends a study from Fyson et al.² that allocated negative emissions only, based on responsibility or capability. Fyson et al. explain that obligations to deliver negative emissions require uncertain technologies made necessary because of insufficient global emissions reductions to date. That study alone could not be used to inform economy-wide emissions targets, and thus not assess the ambition of NDCs, as it only allocated negative emissions and “assume[d] that positive emissions follow least-cost pathways (that is, no equity principle is applied to gross emissions)”².

The figures and captions appear in a mess. The same figures appear on multiple pages, sometimes with and sometimes without captions.

Indeed, our apologies for that. I believe that the figure referencing system of word is not well handled by the pdf conversion tool on the journal’s platform.

The discussion has no structure, which makes it difficult to follow. There is no conclusion.

We have thoroughly revised the discussion that now focusses on 1) how continuity assumptions are present in the literature, 2) how our combination of equity principles compares to the literature, 3) how our allocation results compare to the literature. Note that Nature Communication’s format does not allow for a conclusion section. The last paragraph was revised and clearly introduced as a conclusion.

We introduce the discussion section with:

“Here we discuss how this study’s modelling choices compare to the literature regarding the continuity assumption and the combination of equity principles. Then, we compare results.”

The methods section appears to contain considerable overlap with the main text (partly reflecting that the main text is very technical).

Thank you for the thorough review. We removed content from the manuscript that already appeared in the methods. Additionally, we moved some content to the methods.

Re. my suggestion to “spend some more words on the criticism past studies have received for including transition periods (e.g., Kartha et al (2018)) and the weak ethical basis for such periods (e.g., Flerbaey et al (2014)). On the other hand, possible drawbacks of removing transition periods could also be discussed. For example, this means that reduced and avoided emissions are treated symmetrically, but the costs of reducing emissions are likely larger than the costs of avoiding future increases. The argument in the paper and in the literature against transition periods relies on emissions trading (ITMOs), but the imperfections of current institutions for this should be mentioned. Perhaps also refer to Knight (2013) for a defense of

moderate grandfathering.”

The authors responded “Thank you for the reference. We added it in the following sentence as follows with references to Fleurbaey and Knight:

“Considering continuous emissions trajectories that look realistic²² implies that present-day levels of domestic emissions are an acceptable starting point in terms of mitigation effort with a utilitarian perspective²⁵.” ‘

I would have liked to see a more engagement with the suggestion than just adding one sentence.

Thank you for standing for your point.

In the fair-share literature, the allocation of emissions space does not distinguish treated reduced and avoided emissions similarly, regardless of whether a transition period is used or not. A transition period does not account for reduction potential, as opposed to IAM scenarios for example. For this reason, we do not find it possible to relate the inclusion of a transition period to the distinction between reduced and avoided emissions. However, the allocation of emissions space into fair-shares may result in ‘hot air’ as the emissions allocation of a country may exceed the emissions of its business as usual scenario. While a transition period mitigated this effect, it does not avoid hot air. Even a pure grandfathering approach could theoretically result in hot air allocations.

We raise the issue of hot air, made more visible through discontinuous allocations in the following paragraph:

“The near-term allocation of some countries, mostly sub-Saharan countries, may exceed their current emissions and business-as-usual trajectory beyond 2030, implying mitigation efforts only later³. However, staying within such decreasing allocations beyond 2030 implies immediate investments, possibly with international support. International support can enable recipient countries to implement mitigation measures in line with the underlying global socio-economic scenario in the near term. Approach 2 uses allocations inversely proportional to GDP per capita^{4,5} (see methods), resulting in high emissions allocations compared to current emissions and allocations based on business-as-usual trajectories^{3,6} for countries with very low GDP per capita (e.g., Ethiopia, Democratic Republic of Congo). These allocations theoretically imply financial transfers that may go beyond needs-based considerations and contribute to poverty reduction through climate action⁷.

Regarding the arguments raised by Knight for the justification of a moderate grandfathering, we discuss the realist justification and the utilitarian justification, with references to Knight and Fleurbaey. Indeed, all international agreements reflect some inertia that give a realist justification to grandfathering. However, we focus on the modelling of equity-based allocations to inform processes not limited to negotiating agreements, and that include climate litigation. We seek to rely directly on concepts of international law and the Paris Agreement that do not support grandfathering. We hope that the following modifications clarify this point:

“The legacy influence of current emissions levels on near-term emissions allocations is described here as a ‘grandfathering’ effect⁸. This grandfathering influence on equity-based emissions allocation is strongest in the near term and increasingly affects

the ambition assessment of NDCs in 2030. As we near 2030, a given NDC's emissions target will be closer and closer to a continuous emissions allocation that is iteratively updated (Figure 1). The grandfathering allocation is criticized for its lack of ethical basis⁹⁻¹¹ and has been shown to penalize the poorest countries¹² as it preserves a status-quo, including current inequalities. Prior to the Paris Agreement, a study highlighted the value of a 'moderate grandfathering'¹³, from a political theory perspective, with a realist justification for negotiations and a utilitarian justification. Indeed, the pledges of many high-emitters only align with a grandfathering allocation⁴. However, the IPCC has highlighted the need for a fair distribution of mitigation efforts, excluding grandfathering, in order to achieve an effective global agreement on emissions reductions^{1,10}. Likewise, recent reports of scientific advisory bodies have disapproved grandfathering when presenting fair-share emissions allocation^{14,15}. The Paris Agreement now requires NDCs of the 'highest possible ambition' that reflect equity. A recent study⁸ described grandfathering allocations as not in line with international law. It identified that all continuous allocations entail elements of grandfathering but did not offer a solution."

Regarding the justification of utilitarianism, we added the following sentence in the next paragraph:

"The utilitarian justification¹³ for a moderate grandfathering relies domestic mitigation costs and is no longer relevant when allocations can be traded to achieve a globally cost-effective pathway¹⁰."

This interpretation is based on p. 417 of Knight's analysis¹³ stating: "On welfarist views such as utilitarianism, the relevant costs are welfare costs, rather than the monetary marginal abatement costs familiar from economics. According to utilitarianism, welfare costs are more important the greater they are. It might be claimed that high emitters face high marginal abatement costs. The marginal abatement cost is the welfare cost of one extra unit of emissions reduction. Thus, the main claim of the marginal cost argument is: one extra unit of emission reductions from a baseline of actual prior emissions decreases welfare to a greater extent where it is assigned to a high emitter than where it is assigned to a low emitter. If this is correct, utilitarianism will maintain that high emitters have greater entitlements, as this will save them – and the global economy – from the severe effects of deeper cuts (cf. Wesley and Peterson 1999, p. 186)."

We also highlight the possible important shortcoming of Article 6:

"As a novel mechanism, the international trading of mitigation outcomes raises implementation issues regarding the additionality of the finance and of the funded mitigation measures. Scrutiny will be needed to ensure the integrity of mitigation measures under Article 6 whose implementation rules were just adopted at COP29, with safeguards on human rights and the additionality of emissions reductions¹⁶⁻¹⁸."

Reviewer #2 (Remarks to the Author):

Thank you for the opportunity to review the revised manuscript.

I have reviewed the revisions made by the authors in response to my previous round of review comments. I am satisfied with these revisions.

Thank you for your time and reviews.

References:

1. Clarke, L. *et al.* Chapter 6 Assessing Transformation Pathways. In: *IPCC AR5 WGIII*. 413–510 (2014).
2. Fyson, C. L., Baur, S., Gidden, M. & Schleussner, C. F. Fair-share carbon dioxide removal increases major emitter responsibility. *Nature Climate Change* **10**, 836–841 (2020).
3. van den Berg, N. J. *et al.* Implications of various effort-sharing approaches for national carbon budgets and emission pathways. *Climatic Change* **162**, 1805–1822 (2020).
4. Robiou du Pont, Y. *et al.* Equitable mitigation to achieve the Paris Agreement goals. *Nature Climate Change* **7**, 38–43 (2017).
5. Jacoby, H. D., Babiker, M. H., Paltsev, S. & Reilly, J. M. *Sharing the Burden of GHG Reductions*. MIT Joint Program on the Science and Policy of Global Change 1–28 <https://globalchange.mit.edu/publication/14428> (2008).
6. Holz, C., Kartha, S. & Athanasiou, T. Fairly sharing 1.5: national fair shares of a 1.5 °C-compliant global mitigation effort. *International Environmental Agreements: Politics, Law and Economics* **18**, 117–134 (2017).
7. Budolfson, M. B. *et al.* Utilitarian benchmarks for emissions and pledges promote equity, climate and development. *Nature Climate Change* **11**, 827–833 (2021).
8. Rajamani, L. *et al.* National ‘fair shares’ in reducing greenhouse gas emissions within the principled framework of international environmental law. *Climate Policy* **21**, 1–22 (2021).
9. Caney, S. Justice and the distribution of greenhouse gas emissions. *Journal of Global Ethics* **5**, 125–146 (2009).
10. Fleurbaey, M. *et al.* Chapter 4. Sustainable Development and Equity. In: *Climate Change 2014: Mitigation of Climate Change. Contribution of Working Group III to the Fifth Assessment Report of the Intergovernmental Panel on Climate Change* 283–350 (2014).
11. Dooley, K. *et al.* Ethical choices behind quantifications of fair contributions under the Paris Agreement. *Nature Climate Change* **11**, (2021).
12. Kartha, S. *et al.* Cascading biases against poorer countries. *Nature Climate Change* **8**, 348–349 (2018).
13. Knight, C. What is grandfathering? *Environmental Politics* (2013) doi:10.1080/09644016.2012.740937.

14. European Scientific Advisory Board on Climate Change. *Scientific Advice for the Determination of an EU-Wide 2040 Climate Target and a Greenhouse Gas Budget for 2030-2050*. <https://doi.org/10.2800/609405> (2023).
15. *A Justified Ceiling to Germany's CO₂ Emissions: Questions and Answers on Its CO₂ Budget*. https://www.umweltrat.de/SharedDocs/Downloads/EN/04_Statements/2020_2024/2022_09_The_CO2_budget_approach.html (2022).
16. COP29: Key outcomes agreed at the UN climate talks in Baku. *Carbon Brief* <https://www.carbonbrief.org/cop29-key-outcomes-agreed-at-the-un-climate-talks-in-baku/#6> (2024).
17. Songwe, V., Stern, N. & Bhattacharya, A. *Finance for Climate Action: Scaling up Investment for Climate and Development*. (2022).
18. Haynes, R. & Benjamin, L. Ambition-raising and ambition-reducing features of the Paris Agreement. in *Research Handbook on the Law of the Paris Agreement* (ed. Zahar, A.) 126–142 (Edward Elgar Publishing, 2024). doi:10.4337/9781800886742.00012.